# The primary mechanism for highly potent inhibition of HIV-1 maturation by lenacapavir

Szu-Wei Huang[1]ʘ*, Lorenzo Briganti[1]ʘ, Arun S. Annamalai[1]ʘ, Juliet Greenwood[2], Nikoloz Shkriabai[1], Reed Haney[1], Michael L. Armstrong[3], Michael F. Wempe[3]¤, Satya Prakash Singh[4], Ashwanth C. Francis[4], Alan N. Engelman[2,5], Mamuka Kvaratskhelia[1]*

1 Division of Infectious Diseases, University of Colorado Anschutz Medical Campus School of Medicine, Aurora, Colorado, United States of America, 2 Department of Cancer Immunology and Virology, Dana-Farber Cancer Institute, Boston, Massachusetts, United States of America, 3 Department of Pharmaceutical Sciences, University of Colorado Anschutz Medical Campus Skaggs School of Pharmacy and Pharmaceutical Sciences, Aurora, Colorado, United States of America, 4 Department of Biological Science, Florida State University, Tallahassee, Florida, United States of America, 5 Department of Medicine, Harvard Medical School, Boston, Massachusetts, United States of America

ʘ These authors contributed equally to this work.
¤ Current address: Department of Biological and Physical Sciences, Kentucky State University, Frankfort, Kentucky, USA
* szu-wei.huang@cuanschutz.edu (S-WH); mamuka.kvaratskhelia@cuanschutz.edu (MK)

## Abstract

Lenacapavir (LEN) is a highly potent, long-acting antiretroviral medication for treating people infected with muti-drug-resistant HIV-1 phenotypes. The inhibitor targets multifaceted functions of the viral capsid protein (CA) during HIV-1 replication. Previous studies have mainly focused on elucidating LEN's mode of action during viral ingress. Additionally, the inhibitor has been shown to interfere with mature capsid assembly during viral egress. However, the mechanism for how LEN affects HIV-1 maturation is unknown. Here, we show that pharmacologically relevant LEN concentrations do not impair proteolytic processing of Gag in virions. Instead, we have elucidated the primary mechanism for highly potent inhibition of HIV-1 maturation by sub-stoichiometric LEN:CA ratios. The inhibitor exerts opposing effects on formation of CA pentamers versus hexamers, the key capsomere intermediates in mature capsid assembly. LEN impairs formation of pentamers, whereas it induces assembly of hexameric lattices by imposing an opened CA conformation and stabilizing a dimeric form of CA. Consequently, LEN treatment results in morphologically atypical virus particles containing malformed, hyper-stable CA assemblies, which fail to infect target cells. Moreover, we have uncovered an inverse correlation between inhibitor potency and CA levels in cell culture assays, which accounts for LEN's ability to potently (with picomolar $EC_{50}$ values) inhibit HIV-1 maturation at clinically relevant drug concentrations.

## Author summary

Lenacapavir (LEN) is used to treat adults with multi-drug-resistant HIV-1 infection. LEN is the first in class, highly potent and long-acting antiretroviral, which works by a

**Data availability statement:** The coordinates of the crystal structure of NTDLEN are available at the Protein Data Bank (PDB) under accession number 8V23. All other data are included in the manuscript and the Supporting information files.

**Funding:** This work was supported by the National Institute of Allergy and Infectious Diseases grants T32 AI150547 to SWH, R01 AI157802 and R01 AI162665 to MK, U54 AI170855 to ACF, ANE and MK, U54 AI170791 to ANE, and R21 AI174879 to ACF. The funders had no role in study design, data collection and analysis, decision to publish, or preparation of the manuscript.

unique mechanism of targeting the viral capsid protein. LEN is administered through subcutaneous injections twice-yearly in combination with other HIV-1 medications. The long-acting antiviral activity relies on the ability of sub-nanomolar LEN concentrations to inhibit HIV-1 replication. Accordingly, cell culture-based mechanistic studies are expected to focus on elucidating highly potent antiviral modes of action of LEN. We and others have previously reported the primary mechanism of inhibition of HIV-1 ingrees by LEN. Our studies here have deciphered previously undescribed mechanistic and structural bases for a highly potent antiviral activity of LEN during viral egress. Specifically, we show that clinically relevant LEN concentrations induce aberrant capsid protein assembly during virus maturation and consequently yield non-infectious HIV-1 paricles. These findings will inform clinical applications of LEN as a potent HIV-1 maturation inhibitor and aid the development of second-generation inhibitors targeting assembly of the mature viral capsid.

## Introduction

The mature HIV-1 capsid, which houses the viral RNA genome, nucleocapsid protein, and key viral enzymes reverse transcriptase and integrase needed for virus replication, is assembled during virion morphogenesis. Following the proteolytic cleavage of the Gag polyprotein, newly released capsid proteins (CAs) assemble in the presence of the cellular metabolite inositol hexakisphosphate (IP6) to form a conical structure [1–3]. IP6 binds to the central cavity of key CA pentamer and hexamer assembly intermediates by engaging with Arg18 and Lys25 [1,4–6]. The main body of the mature capsid contains hexameric lattices, whereas seven and five pentamers positioned at the wider and narrower ends ensure that the conical structure is fully closed [5–9]. The mature capsid is the principal interface with the cellular milieu, as it encounters both host-dependency cofactors and antiviral restriction factors during trafficking of HIV-1 cores across the cytoplasm, through the nuclear pore complex and inside the nucleus [10,11].

HIV-1 CA is an important therapeutic target. The first-in-class HIV-1 CA targeting antiretroviral lenacapavir (LEN, Gilead Sciences) is used to treat individuals infected with muti-drug-resistant HIV-1 phenotypes [12]. Structural studies have revealed that LEN binds a CA hexamer by targeting a hydrophobic pocket formed by two adjoining CA subunits [13,14]. The inhibitor primarily engages with the N-terminal domain (NTD) of one CA subunit through extensive hydrophobic and electrostatic interactions. Furthermore, LEN establishes additional hydrogen bonding interactions with the C-terminal domain (CTD) of an adjoining CA subunit. Consistent with these structural findings, LEN bound to cross-linked CA hexamers with a higher affinity ($K_D$ of ~ 200 pM) compared to CA monomers ($K_D$ of ~ 2 nM) [13].

Mechanistic experiments have uncovered that LEN targets multiple steps of HIV-1 replication [13–17]. The inhibitor blocks HIV-1 nuclear import and integration with sub-nM $EC_{50}$ values by hyper-stabilizing the pre-formed mature capsid [14,16]. At substantially higher concentrations (~ 100 nM), LEN can break up the conical structure during virion ingress and block reverse transcription [18]. While previous studies have primarily focused on elucidating the LEN mode of action during early steps of HIV-1 replication, it has also been reported that the inhibitor impairs late steps of HIV-1 replication [13]. Specifically, LEN induces aberrant capsid formation in virions and *in vitro* [13,19]. However, the underlying mechanistic and structural bases for LEN antiviral activity during virion morphogenesis are unknown.

Here, we uncover the primary antiviral mechanism of LEN action during virion morphogenesis. LEN exhibits opposing effects on the formation of pentamers and hexamers, which are the key capsomere assembly intermediates needed to build the mature conical capsid. LEN

impairs formation of pentamers, whereas the inhibitor promotes formation of hexameric lattices by inducing an opened protein conformation and stabilizing a dimeric form of CA. The LEN-induced disbalance of pentamer versus hexamer formation in turn leads to improper assembly of capsid lattices in the presence of IP6. Consequently, the virions produced in the presence of LEN contained malformed, hyper-stable CA assemblies and were unable to productively infect target cells. Furthermore, we have uncovered an inverse correlation between LEN potency and p24 levels. LEN inhibited production of infectious virions with $EC_{50}$ of 60 pM at the lowest p24 levels assayed here. These findings have clinical implications for the use of LEN as a highly potent inhibitor of HIV-1 maturation.

## Results

### LEN potency during late steps of HIV-1 replication is strongly influenced by CA levels

Previous studies that determined LEN $EC_{50}$ values during late steps of HIV-1 replication used pseudotyped viruses, which yielded substantially reduced levels of virions compared to wild-type (WT) full-length HIV-1 [13]. It is unknown whether varying levels of virion production could influence LEN $EC_{50}$ values. Furthermore, in published studies [13], excess inhibitor was not removed from the virions prepared in the presence of LEN, thereby raising concerns that carry over amounts of inhibitor into target cells could influence resulting $EC_{50}$ measurements.

Here we examined how different levels of virion production affected the antiviral potency of LEN during late steps of HIV-1 replication. For this, HEK293T cells were transfected with increasing concentrations of WT full-length pNL4.3 plasmid DNA. The total amount of transfected DNA was kept constant (2 μg) throughout different experiments by supplementing the HIV-1 expressing plasmid with empty vector DNA. LEN was added to the cell culture 6 h post transfection. Resulting levels of HIV-1 virions were evaluated by the p24 ELISA, which increased with increasing levels of pNL4.3 (S1 Table). To determine LEN $EC_{50}$ values, excess inhibitor was removed from virions by centrifuging the drug-treated particles through 20% sucrose cushions. The infectivity of these virions was then measured in target TZM-bl cells.

The results in Fig 1 show that LEN potency concomitantly and markedly decreased (Fig 1A–1D) with increasing levels of p24 (S1 Table). Specifically, LEN $EC_{50}$ values changed > 100-fold, from ~ 60 pM to ~ 6.7 nM, with the lowest and highest levels of p24, respectively, tested here. The detection limits of HIV-1 infectivity assays did not permit us to test LEN potency at even lower, more clinically relevant levels of p24 (see Discussion). Yet, the inverse correlation between LEN potency and p24 levels observed in our experiments (Fig 1) strongly suggests that the inhibitor can potently (with pM $EC_{50}$ values) inhibit late steps of HIV-1 replication in the clinical setting.

In control experiments, $EC_{50}$ values of another late-stage HIV-1 replication inhibitor, Darunavir (DRV), which targets the viral protease (PR) catalytic activity, were only modestly (~ 2-fold) affected under identical assay conditions (S1 Fig). We suggest that DRV $EC_{50}$ values are driven by its binding affinity to viral PR rather than the drug to target ratios. Taken together, our findings reveal that LEN potency is significantly influenced by the CA levels in virus producer cells and suggest the importance of LEN:CA ratios for the mode of action of the inhibitor.

### LEN does not inhibit proteolytic processing of Gag during virion maturation

Previous studies with pseudotyped HIV-1 uncovered two distinct activities of LEN in virus producer cells: i) the virions prepared in the presence of LEN exhibited malformed cores that were morphologically distinct from mature and immature particles, and ii) LEN treatments of

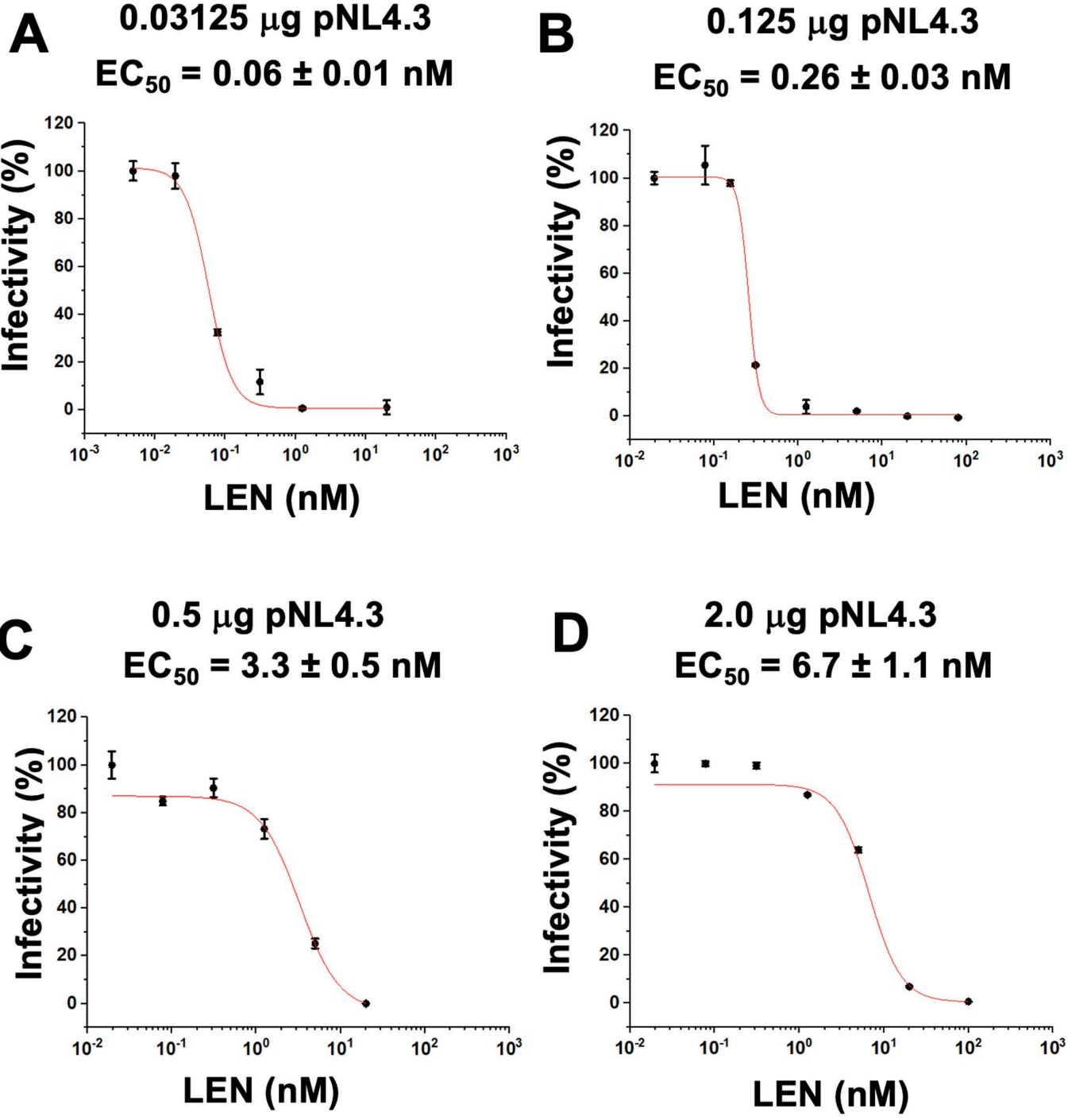

**Fig 1. LEN potency is influenced by p24 levels.** (A–D) HIV-1 virions were produced by transfecting HEK293T cells (producer cells) with indicated concentrations of full-length WT pNL4.3. Indicated concentrations of LEN or DMSO control were added to HEK293T cells, and viruses isolated by ultracentrifugation through 20% sucrose cushions were used to infect HeLa TZM-bl cells (Target cells). After 48 h of infection, luciferase activity was measured to determine $EC_{50}$ values of LEN. The averaged data (+ / − SD) from three independent experiments are shown.

$\geq 10$ nM reduced intracellular Gag levels [13]. Here we have reanalyzed these activities of LEN in the context of full-length WT HIV-1$_{NL4.3}$. Under identical assay conditions of transfecting 2 µg pNL4.3, LEN $EC_{50}$ of ~ 6.7 nM impaired virion infectivity (Fig 1D), whereas the reduction

of intracellular Gag levels was observed at > 75 nM LEN (S2 Fig). The mechanism by which LEN reduces Gag levels in virus producer cells is unknown and was not further investigated here. Instead, our studies focused on the more potent antiviral activity of LEN during virion morphogenesis.

To examine the inhibitor effects on virion morphology, the virus producer cells were treated with 30 nM and 60 nM LEN (Fig 2). These inhibitor concentrations were chosen to elicit detectable drug effects on assembly of the mature capsid at levels of virion production needed for transmission electron microscopy (TEM) analysis and to avoid substantial reduction of intracellular Gag levels observed at > 75 nM LEN (S2 Fig). The analysis of virions by TEM revealed three broad classes of mature, immature, and atypical phenotypes (Fig 2A). The atypical species included rod-shaped cores, empty cores, multiple cores, and multiple densities, which could not always be clearly delineated from one another. Therefore, they were grouped as cumulative totals of defective or atypical virions. Viruses made in the absence of LEN were morphologically ~ 81.8% mature, 3.6% immature, and 14.6% atypical (Fig 2B). The

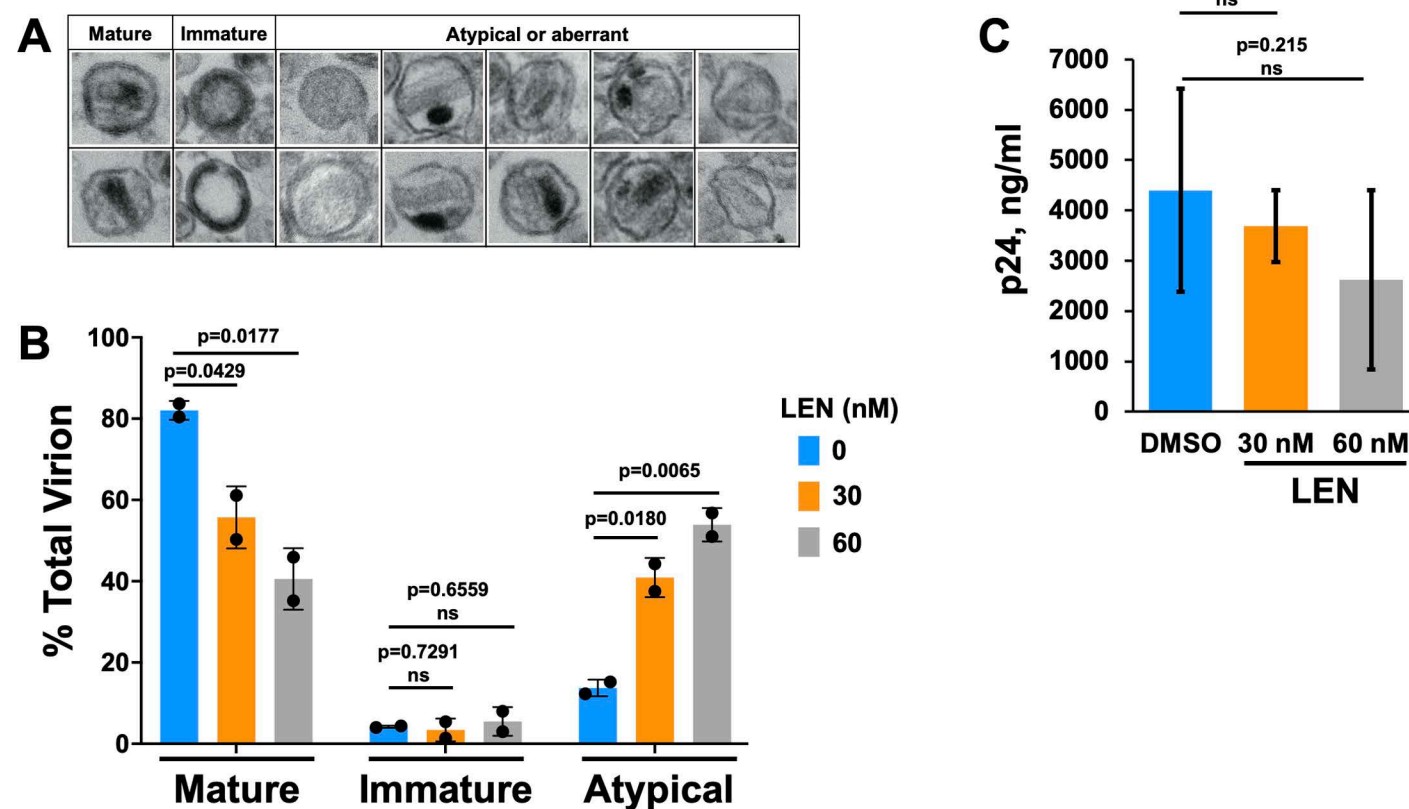

**Fig 2. LEN treatments yield virions with malformed capsids.** (A) Representative micrographs of virions used during morphological classification. For each experimental condition, > 200 virions were counted. Across cumulative experiments (n = 3 DMSO treatment; n = 2 for LEN treatments), > 520 virions were counted in total. Virions with visibly electron-dense outer membranes were classified as: mature, containing a single circular or conical electron-dense region; immature, containing outer, comparatively thick toroidal or semi-hemispheric electron density; the following were grouped as atypical or aberrant: empty, comparative lack of luminal contents; rod, electron-dense or electron-lucent core lacking obvious conical shape; multiple cores, > 1 conical, circular, or rod-like capsid shape that was electron-dense or lucent; multiple densities; > 1 distinct electron-dense region wherein at least one of the densities could not be readily attributed to a core structure; empty core, roughly conical shapes that lacked associated electron density. (B) Quantitation of virions belonging to indicated morphological categories (average +/ − SD for n = 2−3 independent experiments). (C) p24 concentrations were determined by p24 ELISA assay after DMSO or LEN treatments (average +/− SD for n = 2−4 independent experiments).

quantitative results in Fig 2C show that LEN treatments did not substantially reduce overall levels of p24, yet the inhibitor significantly reduced mature particle yield with concomitant increases of atypical particles (Fig 2B). At the highest LEN concentration tested (60 nM), atypical structures outnumbered mature particles.

LEN could inhibit virion maturation by binding Gag and interfering with its proteolytic processing in virions; and/or inducing aberrant assembly of CA in HIV-1 particles. Previous studies [13,15] as well as our findings in Fig 2B demonstrate that LEN treatments do not enhance the levels of immature virions, suggesting that the inhibitor does not affect proteolytic processing of Gag in virions. To further examine this notion, we monitored Gag proteolytic processing products by immunoblotting. In S3A Fig, we added LEN to HEK293T cells 6 h post transfection of the pNL4.3 plasmid and examined virions formed over the ensuing 48 h. For these experiments, we used 2 nM LEN and 0.125 μg pNL4.3 to fully inhibit virion maturation (Fig 1B) yet minimize reduction of intracellular levels of Gag (S2 Fig). The results in S3A Fig show that LEN did not detectably affect proteolytic processing of Gag in virions.

We performed another experiment where 50 nM LEN was added to freshly-washed virus producer cells 30 h post transfection with 2 μg pNL4.3. By this time, ample amounts of intracellular Gag protein were already made. Virus-containing supernatants were monitored at several timepoints over an ensuing 8 h window. The rationale for this experimental design was to separate the ability of LEN to reduce intracellular Gag levels from the inhibitor effects during virion maturation. The results in S3B Fig show that, as expected due to the washing step, the supernatants did not contain detectable levels of virions at the 0 h time point, which in turn indicated that the virions observed at 2 h and subsequent time points were freshly produced in the presence or absence of LEN. 50 nM LEN fully impaired infectivity of the virions (Fig 1D). Yet, Gag proteolytic processing in virions was not detectably affected by 50 nM LEN (S3B Fig).

Next, we conducted complementary biochemical experiments. LEN bound to full length recombinant Gag protein produced from *Escherichia coli* with $K_D$ of > 200 nM (S4 Fig), which is > 3-orders of magnitude higher than LEN's antiviral $EC_{50}$ value of 60 pM (Fig 1). We also examined proteolytic processing products of purified full-length Gag and Gag(Δp6) using recombinant HIV-1 protease (S5 Fig). Again, the kinetics of formation of proteolytic products did not reveal any noticeable differences in the absence vs. the presence of 2 μM LEN (S5 Fig). Taken together, we conclude that LEN does not adversely affect the proteolytic processing of Gag during virion maturation. Accordingly, our future efforts have focused on elucidating how LEN interactions with CA influenced viral capsid assembly.

## Sub-stoichiometric LEN:CA ratios yield morphologically defective capsids

To quantitate interactions of LEN with HIV-1 virions, we have developed a highly sensitive LC-MS/MS-based method (Fig 3A). WT full-length HIV-1$_{NL4.3}$ virions were produced in HEK293T cells in the presence of varying concentrations of LEN and then subjected to ultracentrifugation through 20% sucrose cushions to isolate virions and remove unbound drug. The resulting virion samples were analyzed by three separate assays: i) LC-MS/MS to measure LEN amounts; ii) p24 ELISA to measure CA amounts; and iii) infectivity to determine $EC_{50}$ values of LEN. A LEN standard curve was performed, which yielded the limit of detection (LOD) of 0.42 nM and the limit of quantitation (LOQ) of 1.3 nM with $R^2$ of 0.99. All measured concentrations of LEN bound to HIV-1 virions (Fig 3B and 3C) were above the LOQ and within the linear range of the drug concentration. Representative LC-MS/MS results comparing LEN levels in the presence and absence of HIV-1 virions are shown in Fig 3B. Measurements of the p24 levels in the same samples allowed us to establish the LEN:CA ratios. In turn, these values were used to determine a half effective concentration ($EC_{50}$) of LEN needed to inhibit HIV-1 infection of target TZM-bl cells.

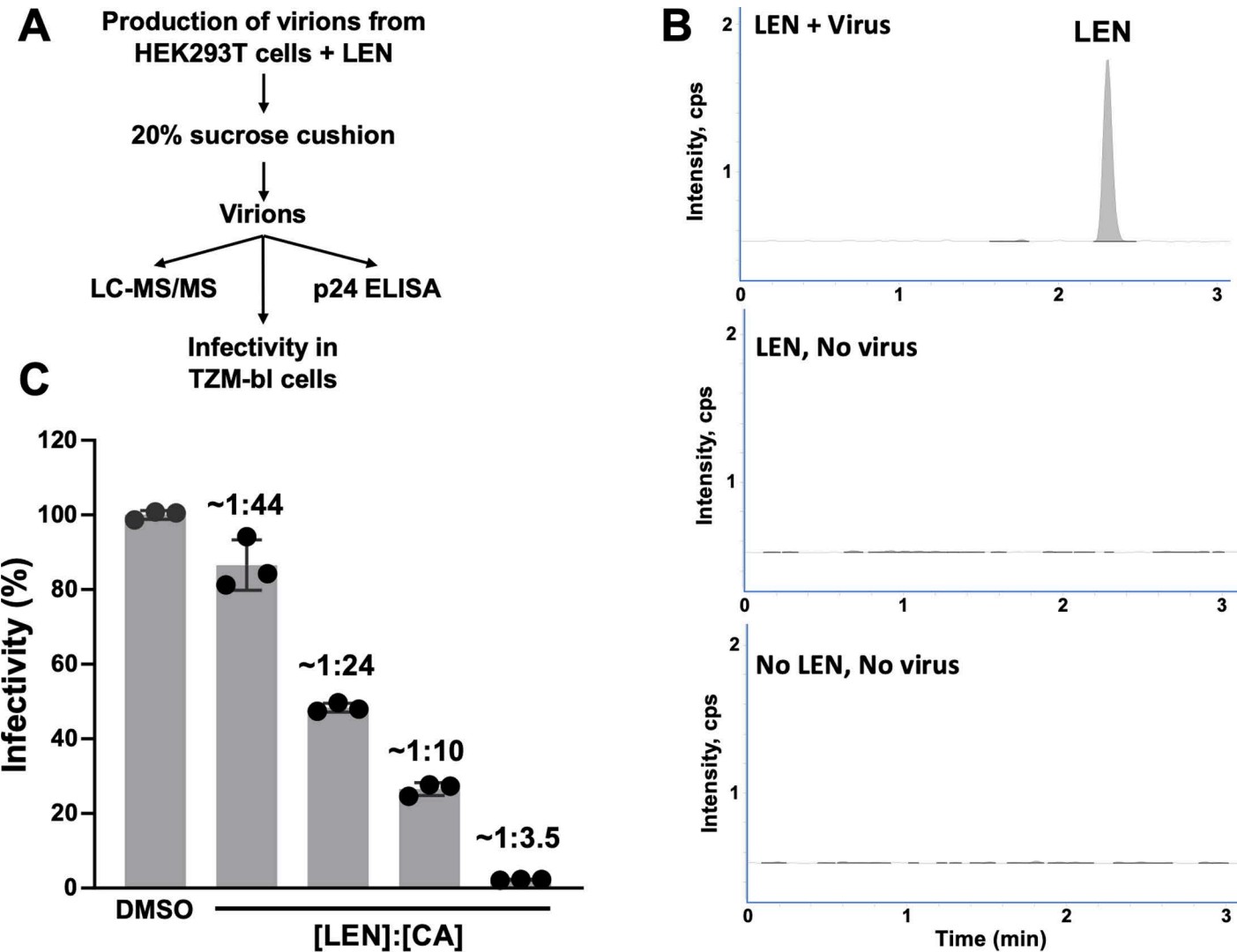

**Fig 3. Sub-stoichiometric LEN:CA ratios inhibit late steps of HIV-1 replication.** (A) The experimental design. The same virus preparation was used for three different assays: 1) LC-MS/MS to determine LEN amounts bound to virions, 2) p24 ELISA to determine CA amounts; and 3) infectivity of the viruses produced in the presence of LEN or DMSO control in HeLa TZM-bl cells. (B) Representative LC-MS/MS results of viruses produced in the presence of LEN (the top panel), the identical concentration of LEN added to HEK293T cells in the absence of pNL4.3 (the middle panel), and the control of the medium without LEN or the virus (the bottom panel). (C) Infectivity of the viruses at indicated [LEN]:[CA] ratios or DMSO control. The [LEN]:[CA] ratios were measured using p24 ELISA and LC-MS/MS results for the same virus preparations.

Strikingly, the inhibitor $EC_{50}$ values were observed at the LEN:CA ratio of ~ 1:24. We note that our p24 measurements indicate total CA levels present in virions. Under baseline assembly conditions, roughly half of the total virion complement of CA assembles into the conical capsid [20,21]. In our assays, LEN was present prior to capsid formation and resulted in malformed CA assemblies. These findings do not allow us to estimate what fraction of total CA protomers are mis-assembled into aberrant oligomers.

Biochemical experiments with purified recombinant protein revealed that LEN promotes aberrant CA assemblies [15,19]. However, the stoichiometry of LEN:CA needed to elicit these effects is not known. Here, we monitored assembly of mature capsid-like particles (mCLPs) *in vitro* using increasing concentrations of LEN. As expected, in control experiments with IP6,

closed conical mCLPs were predominantly observed (Fig 4A). By contrast, LEN in the absence of IP6 resulted in formation of nanotubes (Fig 4B). Simultaneous addition of both IP6 and LEN to monomeric WT CA yielded atypical morphologies which tended to readily clump together (Fig 4C). The atypical morphologies included partly closed structures or other irregularly shaped assemblies (Fig 4C). The quantitative analysis of total LEN vs. CA present in the reactions containing 117 μM CA, 500 μM IP6 and varying concentrations of LEN indicated that mCLP formations were reduced by ~ 50% with concomitant increases of atypical species in the range of the [LEN]:[CA] ratios of 1:16 to 1:32 (Fig 4D).

## LEN specifically blocks formation of pentamers and promotes assembly of hexameric lattices

The underlying mechanistic and structural bases for how LEN induces morphological defects during assembly of mature capsid is not known. To address these important questions, we investigated how addition of LEN to monomeric CA affected formation of pentamers and hexamers, the key capsomere assembly units of the mature HIV-1 capsid. For this, we employed well-studied amino acid substitution mutant proteins CA(N21C/A22C/W184A/M185A) and CA(A14C/E45C/W184A/M185A) to monitor formation of pentamers and hexamers, respectively [22,23]. As expected, in control experiments with DMSO, CA(N21C/

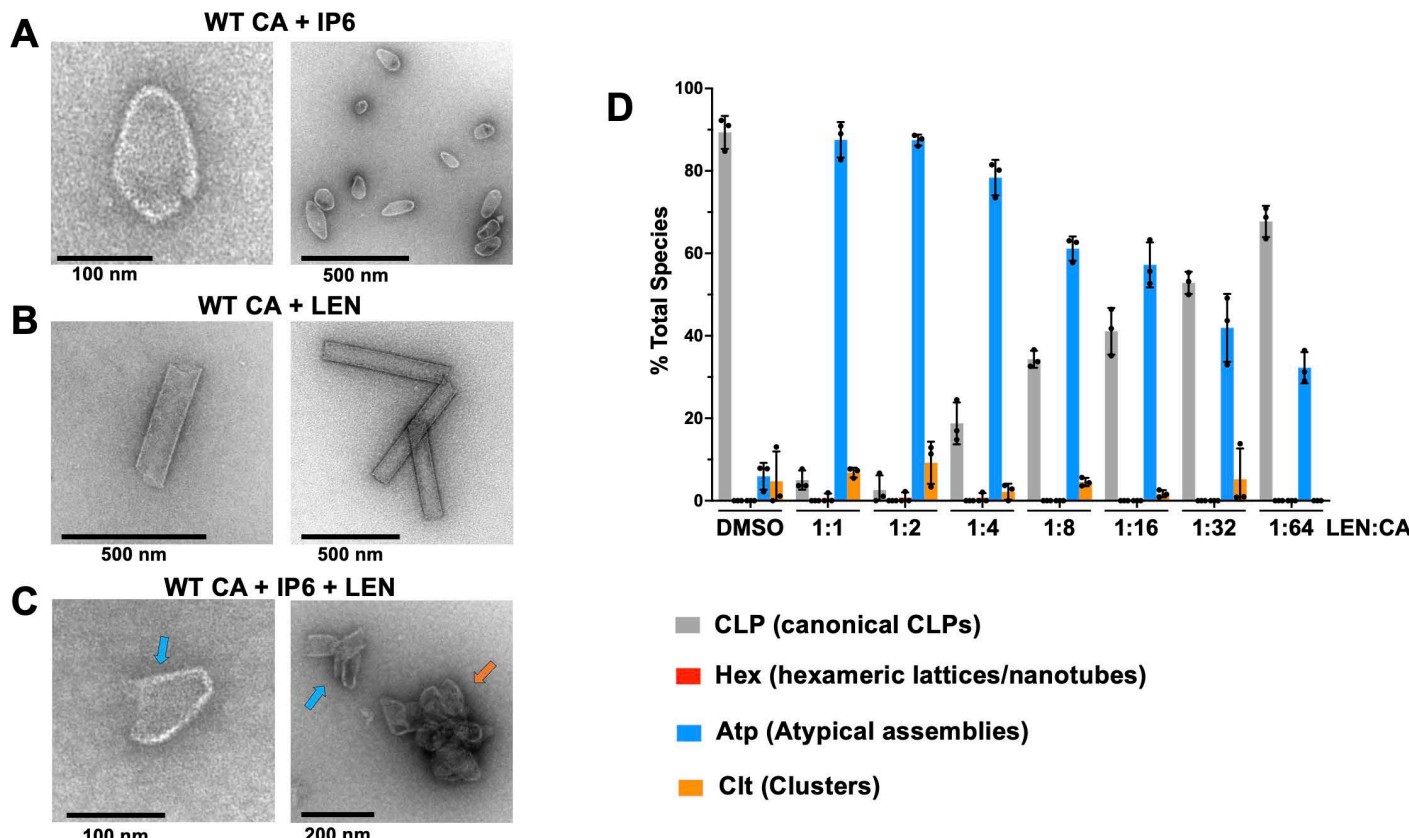

**Fig 4. Sub-stoichiometric LEN:CA ratios impair formation of mCLPs assemblies *in vitro*.** Representative micrographs of WT CA (117 μM) in the presence of (A) 500 μM IP6; (B) 117 μM LEN; and (C) 500 μM IP6 + 117 μM LEN. In vitro assembly reactions were performed in 50 mM MES (pH 6.0) and 40 mM NaCl. (D) Quantification of in vitro assembly products using 117 μM WT CA, 500 μM IP6 and varying amounts of LEN. The [LEN]:[CA] ratios for each reaction is indicated. The averaged results (± SD) from three independent experiments are shown.

A22C/W184A/M185A) and CA(A14C/E45C/W184A/M185A) yielded disulfide-mediated cross-linked pentamers and hexamers, respectively (Fig 5, lanes 3 in A and B). Remarkably, pre-incubation of LEN with monomeric CA specifically and effectively blocked formation of CA pentamers (Fig 5A, lane 4), whereas the inhibitor did not interfere with assembly of CA hexamers (Fig 5B, lane 4).

Previous studies indicated that the structure of the isolated cross-linked pentamer differs from the pentamers found in HIV-1 capsid [24]. The root mean square deviation (RMSD) between the structures of isolated crosslinked vs native pentamers present in the capsid is ~ 2.79 Å. The main difference between these pentamers is in the relative positioning of the CTD with respect to the NTD. We note that in the capsid, the CTD domains of pentamers tightly engage with five surrounding hexamers. Accordingly, it is likely that the orientation of the CTD, which is connected to the NTD through a highly flexible linker, is strongly influenced by interacting hexamers. By contrast, the structure of native isolated pentamers is unknown. Here, we used AlphaFold to generate a model for WT CA pentamers (S6 Fig). Our findings indicate that the predicted structure of isolated WT CA pentamer closely resembles (RMSD of 1.04 Å) the high-resolution structure of the cross-linked pentamer (S6 Fig), lending physiological credence to our findings in Fig 5.

To examine how IP6 and LEN affected WT CA assembly intermediates, we used mass photometry (Fig 6). Unliganded WT CA (Fig 6A), cross-linked pentamers (Fig 6B) and

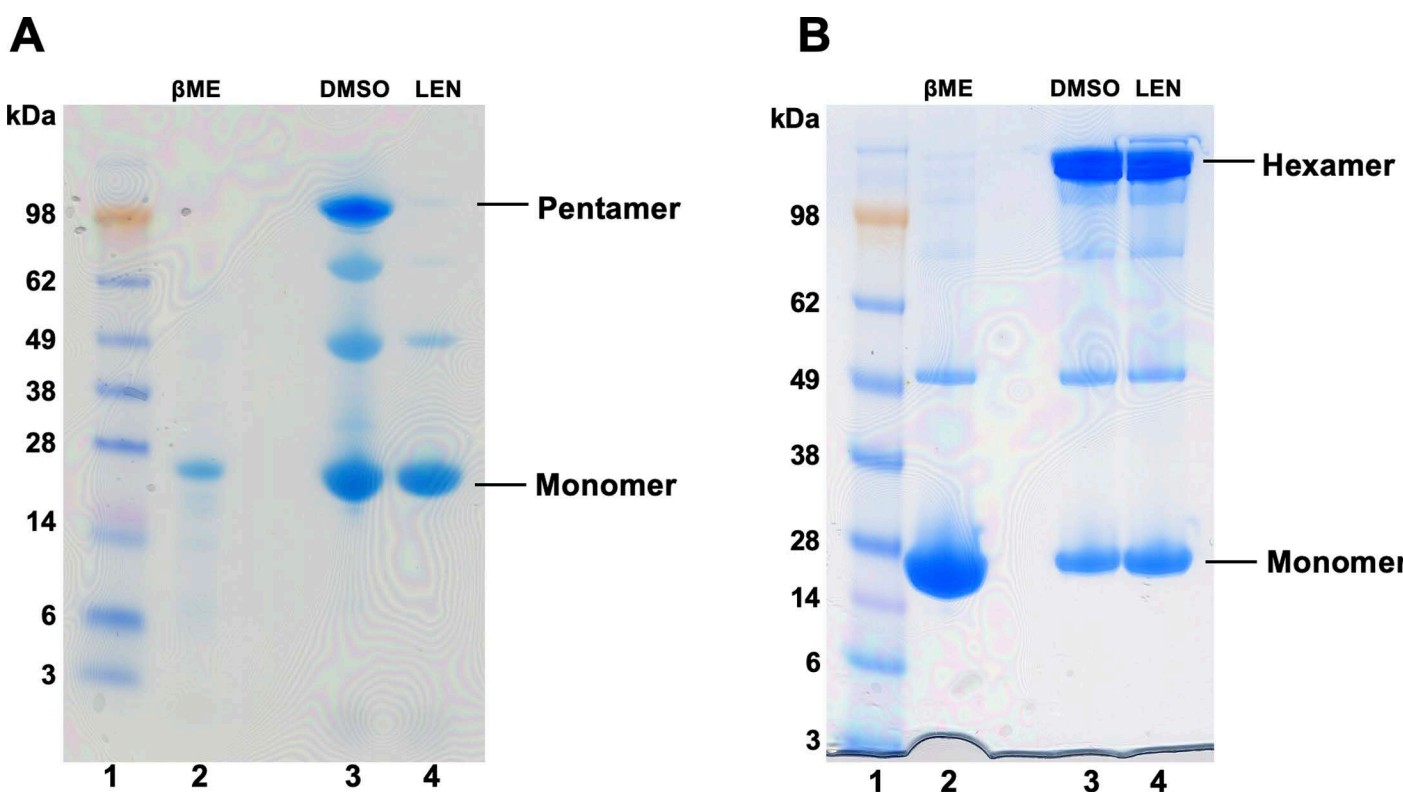

**Fig 5. LEN specifically blocks formation of cross-linked pentamers but not hexamers.** Representative SDS-PAGE images for formation of cross-linked pentamers (A) and cross-linked hexamers (B). 1 μM CA was used in all reactions. Lanes 1 (A and B): molecular weight markers; Lanes 2 (A and B): the control assembly reactions in the presence of 40 mM βME, which inhibits the cross-linking reactions. Lanes 3 (A and B): DMSO was added to monomeric CA in the absence of the reducing agent and then cross-linked products were monitored for pentamers (A) and hexamers (B). Lanes 4 (A and B) 2 μM LEN was added to monomeric CA in the absence of the reducing agent and then cross-linked products were monitored for pentamers (A) and hexamers (B).

cross-linked hexamers (Fig 6C) were examined in control experiments. WT CA in the absence of added ligands was a monomer. The observed molecular weight (MW) of 29 ± 9 kDa (Fig 6A) corresponded with the theoretical MW of ~ 25.4 kDa of a single CA subunit. Measured MWs for cross-linked pentamers and hexamers were 130 ± 12 kDa (theoretical MW= ~ 127 kDa) and 154 ± 14 kDa (theoretical MW = ~ 152), respectively (Fig 6B and 6C).

Striking differences were observed between assembly products of WT CA + IP6 (Fig 6D–6F) vs WT CA + LEN (Fig 6G–6I). The WT CA + IP6 reactions initially (10 min) resulted in a mixture of monomers and dimers as evidenced by the main peak with MW of 30 ± 4 kDa corresponding to monomeric CA (Fig 6A), and the shoulder with MW of 47 ± 7 kDa, which is due likely to CA dimer formation (theoretical MW = ~ 51 kDa). Incubation of WT CA + IP6 for 1 h resulted in a mixture of pentamers and hexamers. The main peak with MW of 119 ± 48 kDa in Fig 6E is within the range of the theoretical MW of ~ 127 kDa for a pentamer, whereas the observed shoulder with MW of 156 ± 14 kDa (Fig 6E) is very close to the theoretical MW of ~ 152 kDa for the hexamer. Longer incubation (16 h) of WT CA + IP6 yielded higher-order CA assembly products with a broad peak of 488 ± 82 kDa (Fig 6F).

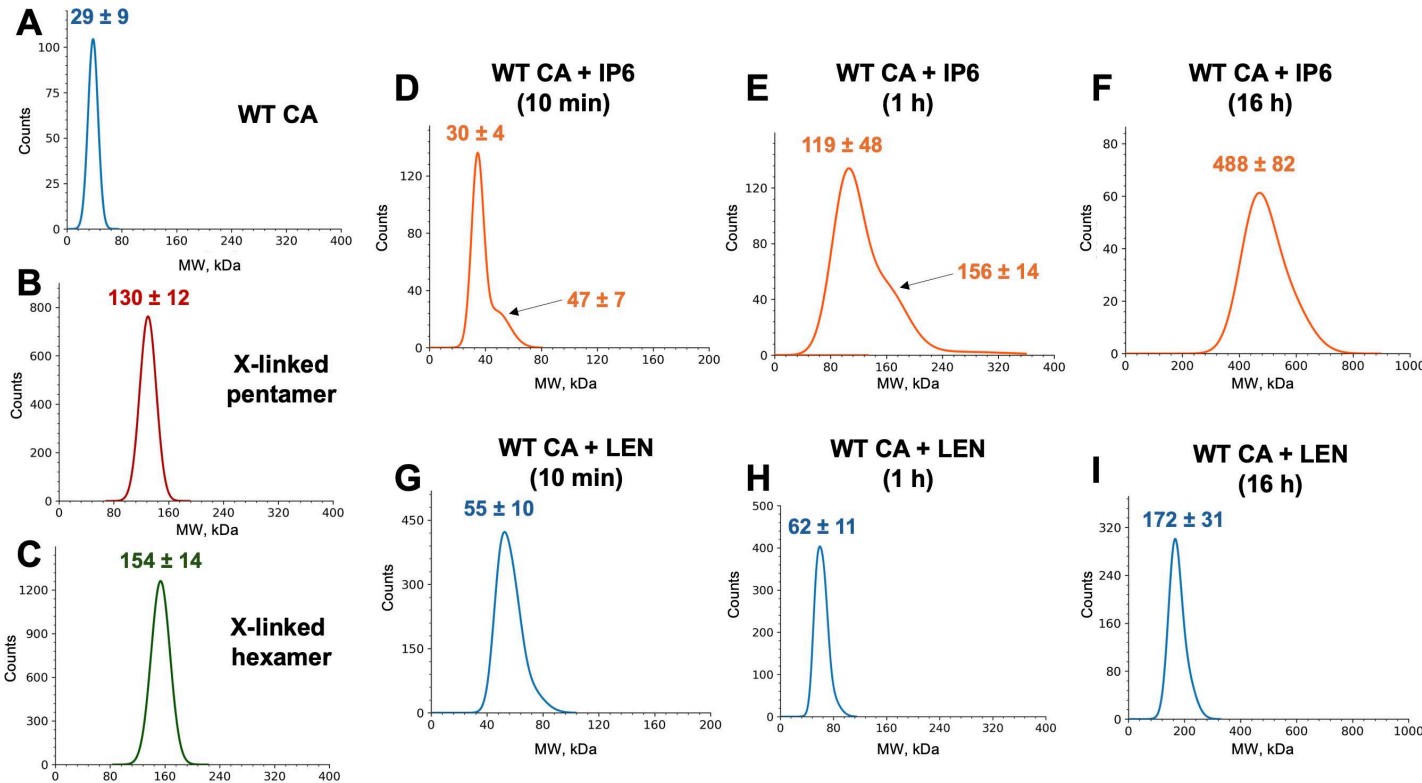

**Fig 6. Mass photometry analysis of WT CA assembly intermediates in the presence of IP6 or LEN.** (A) Unliganded WT CA; The observed peak of 29 ± 9 kDa corresponds to the theoretical MW of a monomer (25.4 kDa). (B) cross-linked CA(N21C/A22C/W184A/M185A) pentamer; (C) cross-linked CA(A14C/E45C/W184A/M185A) hexamer; (D) WT CA + 10 mM IP6 after 10 min incubation. The main peak of 30 ± 4 kDa is close to the theoretical MW of a monomer (25.4 kDa) with a shoulder at 47 ± 7 kDa indicating the formation of a dimer (the theoretical MW of ~ 51 kDa). (E) WT CA + 10 mM IP6 after 1 h incubation. The major peak of 119 ± 48 kDa correlates to the theoretical MW of the pentamer (~ 127 kDa) with a shoulder at 156 ± 14 kDa indicating the presence of a hexamer (the theoretical MW of ~ 152 kDa). (F) WT CA + 10 mM IP6 after 16 h incubation. The major peak of 488 ± 82 kDa indicates higher-order CA assembly products. (G) WT CA + LEN after 10 min incubation. The single symmetrical peak of 55 ± 10 kDa corresponds to the theoretical MW of the CA dimer + LEN is ~ 52 kDa; (H) WT CA + LEN after 1 h incubation. The single symmetrical peak of 62 ± 11 kDa indicates that LEN stabilizes CA dimers; (I) CA + LEN after 16 h incubation. The single symmetrical peak with MW of ~ 172 ± 31 kDa is in the range of the theoretical MW of CA hexamer + LEN (~ 158 kDa).

The WT CA + LEN reactions initially (10 min) yielded a single symmetrical peak with MW of 55 ± 9.6 kDa indicating that the inhibitor stabilized CA dimers (theoretical MW = ~ 51 kDa) (Fig 6G). After 1 h incubation of WT CA + LEN, we still observed the dimeric species (Fig 6H), which subsequently (after 16 h) transitioned into CA hexamers (the observed peak with MW of 172 ± 31 kDa is sufficiently close to the theoretical MW of ~ 159 for CA hexamer + LEN) (Fig 6I).

We note that both WT CA + IP6 and WT CA + LEN reactions are highly dynamic with the observed reaction intermediates being converted into higher order assembly products, and new intermediates being continually formed. Mass photometry cannot detect higher order assembly products such as CLPs or nanotubes, which in turn are readily observed by TEM (Fig 4). Therefore, mass photometry and TEM are complementary techniques for analyzing assembly intermediates and final products. Collectively, these two approaches indicated that the CA + IP6 reactions proceeded through an initial mixture of monomers and dimers followed by pentamer and hexamer intermediates (Fig 6D and 6E) and subsequently resulted in formation of CLPs (Fig 4A). By contrast, LEN initially stabilized CA dimers, then formed hexameric intermediates (Fig 6G–6I) and consequently yielded nanotubes (Fig 4B).

## LEN stabilizes an opened CA conformation conducive to hexamer but not pentamer formation

To understand the structural basis behind observed LEN effects on CA, we determined the X-ray structure of the monomeric N-terminal domain (NTD) of CA in the presence of LEN (NTD$_{LEN}$; Fig 7 and S2 Table). Under our experimental conditions, NTD crystals formed only in the presence, but not in the absence of LEN, indicating a crucial role of the inhibitor in the formation of the protein crystals. The NTD$_{LEN}$ structure was resolved at ~ 2 Å, which revealed inhibitor-induced conformational changes (see below). Yet, we were unable to resolve the density for LEN in the NTD$_{LEN}$ structure. We note that LEN binds to monomeric CA with a lower affinity than pre-formed hexamers [13,15]. More specifically, LEN rapidly dissociates from monomeric CA, whereas the $K_{off}$ rate for the CA$_{Hex}$ + LEN complex is unusually slow [13,15]. Unbound LEN readily precipitates, whereas the drug in the complex with CA$_{Hex}$ is highly soluble [14]. These differential interactions of LEN with CA monomer vs. hexamer could explain as to why we and others were able to obtain crystal structures for highly stable CA$_{Hex}$ + LEN complexes [13–15,25], whereas the inhibitor density was absent from the NTD$_{LEN}$ structure. We reckon that LEN dissociated from monomeric NTD and precipitated during crystal growth, while the inhibitor-induced protein conformation was retained because of lattice contacts during the crystal packing.

We would draw a parallel between our observations and earlier structural studies of the NTD in the presence of the CAP-1 inhibitor [26], which binds monomeric HIV-1 CA with modest affinity and is poorly soluble. Akin to our case, NTD crystallization was CAP-1-dependent [26]. Furthermore, while the density for CAP-1 was not observed in the high-resolution (~ 1.5 Å) structure of the NTD, protein conformational changes induced by CAP-1 were readily apparent [26]. Of note, as discussed below, LEN and CAP-1 render very distinct conformational changes in the protein.

Comparisons of our NTD$_{LEN}$ structure with published structures of unliganded apo-protein (NTD$_{Apo}$), the NTD in the presence of various inhibitors, as well as CA hexamers and pentamers, are telling [6,14,26-29] (Figs 7,S7 and S8). Specifically, we have focused on the protein segment connecting α helices H3 and H4, which was recently shown to influence a molecular switch between pentamers and hexamers [6]. The key players of the switch are the TVGG motif at the C-terminal end of H3 and the gate keeper residue Met66 on H4. In the CA hexamer, the TVGG sequence adopts disordered conformation, and this

segment is substantially distanced from Met66. For example, the distance between Cα of Gly60 and Cε of Met66 is 8.4 Å (S7B Fig). In contrast, the TVGG adopts a helical structure in the CA pentamer and the distance between Cα of Gly60 and Cε of Met66 is 5.3 Å (S7F Fig). Accordingly, here we refer to these two distinct structural organizations of adjoining segments of H3 and H4 seen in hexamers and pentamers as to opened and closed conformations, respectively.

Opened and closed conformations markedly influence LEN binding (S8 Fig). An opened conformation affords sufficient pliability to the Met66 side chain to readily accommodate LEN in the context of CA_Hex [13,15,25] (S7C and S8A Figs). In contrast, the rigid helical structure TVGG in the CA pentamer limits the flexibility of the Met66 side chain, which in turn poses steric hindrance with respect to LEN binding (S8B Fig). Indeed, the recent cryo-EM studies with mCLPs found that LEN was specifically bound to CA hexamers but not pentamers [5].

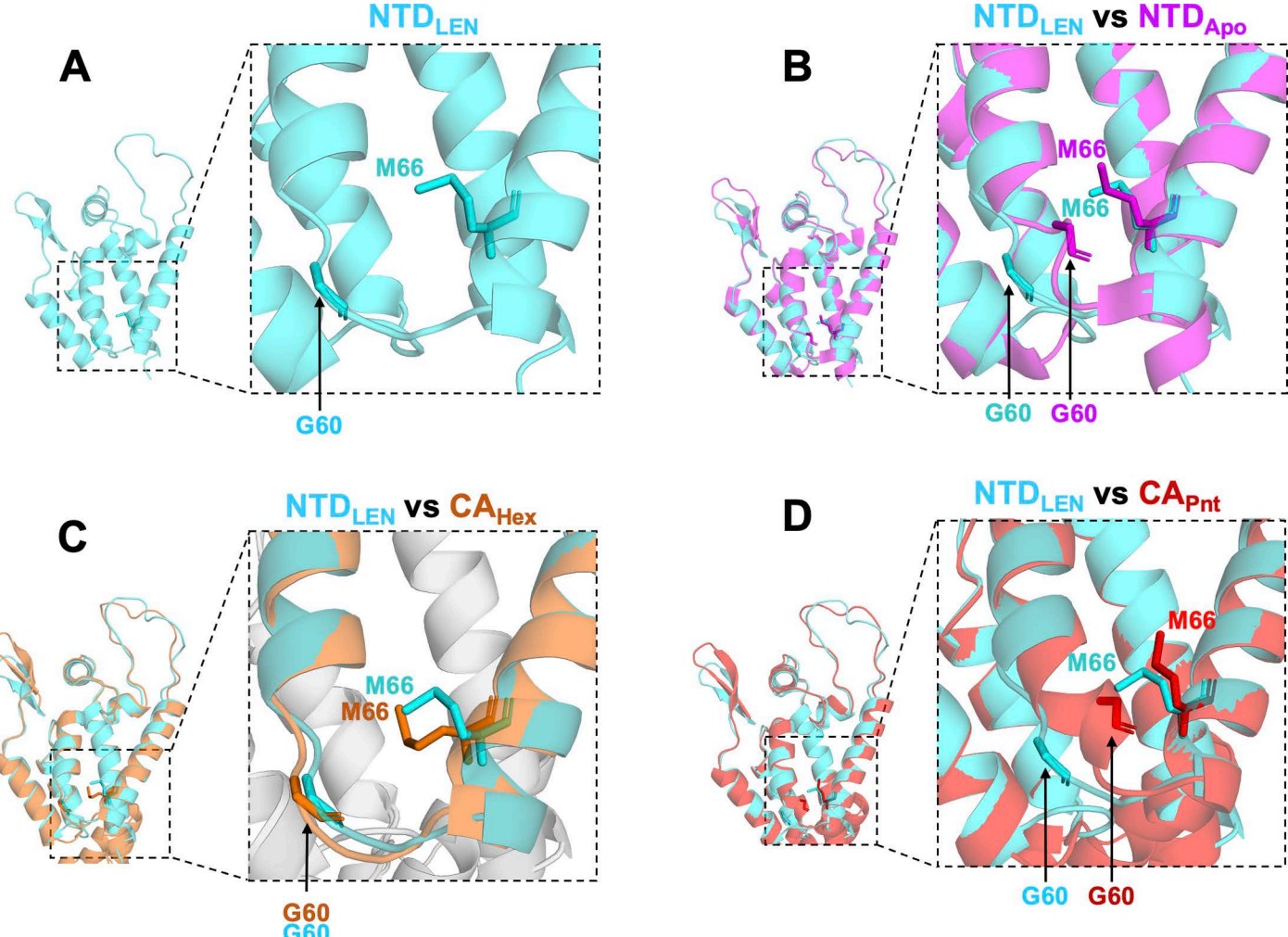

**Fig 7. CA NTD adopts an opened conformation in the presence of LEN.** (A) The X-ray crystal structure of NTD + LEN (NTD_LEN, PDB: 8V23). Side chains of Gly60 and Met66 are indicated to delineate the opened and closed conformations. (B) Our X-ray crystal structure of NTD_LEN (steel blue) superimposed onto the apo NTD (NTD_Apo, magenta, PDB: 5HGK). (C) The X-ray crystal structure of NTD_LEN (steel blue) superimposed onto the native CA hexamer (CA_Hex, orange, PDB: 7URN). (D) X-ray crystal structure of NTD_LEN (steel blue) superimposed onto the native CA pentamer (CA_Pnt, scarlet, PDB: 7URN).

Of note, $NTD_{LEN}$ exhibits an opened conformation (Figs 7 and S7A). In contrast, the $NTD_{Apo}$ adopts a closed conformation even though the TVGG sequence appears to be disordered (Figs 7B, S7E and S8C). Accordingly, LEN can be readily accommodated in the $NTD_{LEN}$ structure (S8A Fig), whereas the inhibitor creates steric hindrance with the Met66 side chain of $NTD_{Apo}$ (S8C Fig). Superimposition of $NTD_{LEN}$ with native $CA_{Hex}$ and $CA_{Pnt}$ reveals that the $NTD_{LEN}$ structure is conducive to hexamer but not pentamer formation (Fig 7C and 7D). We also note that the NTD exhibits an opened conformation in the presence of PF74 (S7D Fig), which similarly to LEN occupies an "upper" hydrophobic CA pocket along H4 [28]. By contrast, the NTD exhibited closed conformations in the presence of CAP-1 and 1F6 inhibitors (S7G and S7H Fig), which, unlike LEN and PF74, are positioned at a "lower" CA pocket in the vicinity of the loop connecting H3 and H4 [26,29]. Collectively, our $NTD_{LEN}$ structure alongside published NTD structures suggest that monomeric apo-CA exhibits conformational flexibility, which is likely to be crucial for its biological function to form both pentamers and hexamers in the presence of the assembly cofactor IP6 [1,6]. The inhibitors that bind at distinct ("upper" or "lower") pockets near H3 and H4 helices limit CA pliability by imposing a preferred (opened or closed) protein conformation. Specifically, LEN induces an opened protein conformation.

## LEN forces CA(M66A) to form hexametric lattices

The M66A substitution in CA has been shown to promote the closed protein conformation as evidenced by the ability of CA(M66A) in the presence of IP6 to predominantly form closed, small spheres made from 12 pentamers and no hexamers [6]. Accordingly, CA(M66A) presented us with a powerful investigational tool to further examine LEN effects on assembly

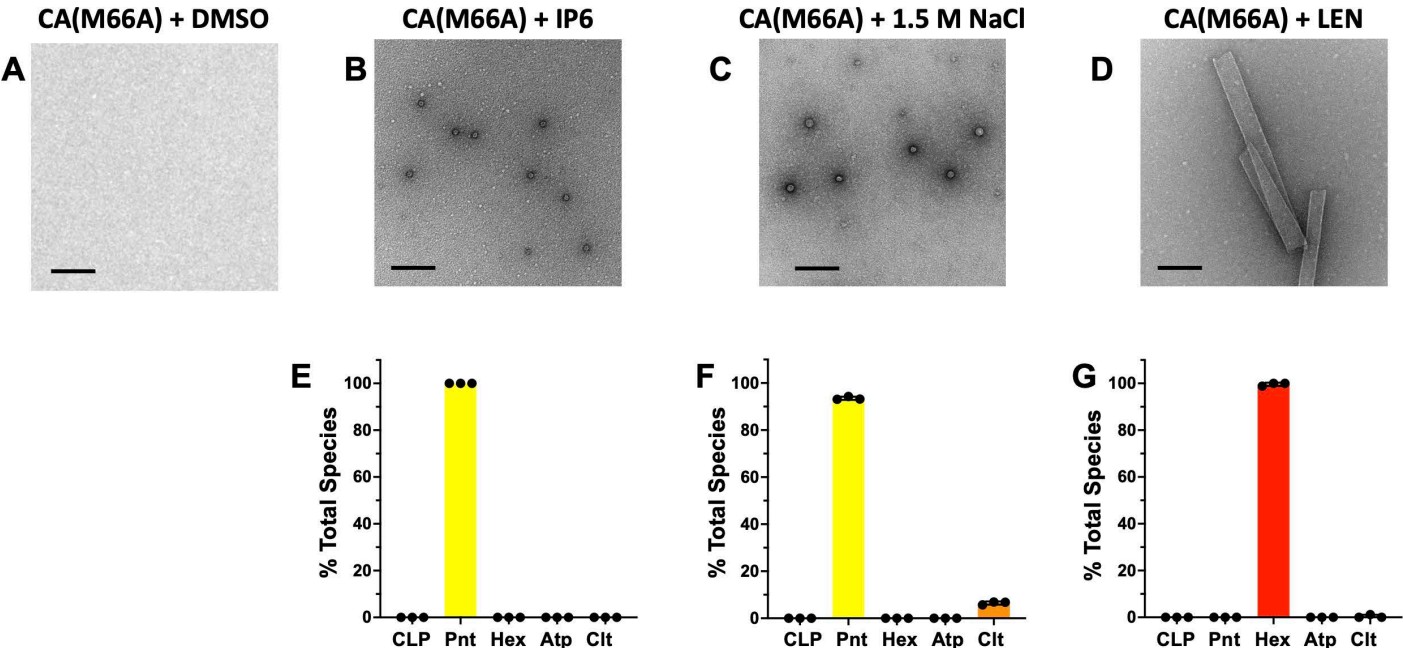

**Fig 8. LEN specifically induces formation of CA(M66A) hexameric lattices.** Representative micrographs of 117 μM CA(M66A) in the presence of (A) DMSO; (B) 10 mM IP6; (C) 1.5M NaCl; or (D) 117 μM LEN. In vitro assembly reactions were performed in 50 mM MES (pH 6.0). (E–G) Quantification of the assembly products shown in B-D. The averaged data (+/− SD) from three independent experiments are shown. CLP, capsid-like particles. Pnt, pentameric spheres. Hex, hexameric lattices/nanotubes. Atp, atypical assemblies. Clt, clusters. Scale bars: 200 nm.

of pentamers vs. hexamers. As expected, incubation of CA(M66A) with IP6 in a low ionic strength buffer resulted in closed, pentameric spheres (Fig 8B). In the presence of 1.5 M NaCl and the absence of IP6, CA(M66A) formed spherical structures which appeared to exhibit a slightly wider diameter than in the presence of IP6 (Fig 8C). Remarkably, in complete contrast from IP6 and 1.5 M NaCl, addition of LEN to CA(M66A) resulted in assembly of exclusively open-ended, hexameric lattices (Fig 8D). Thus, our findings delineate LEN from other small molecule ligands (such as IP6 and NaCl) by its unique mechanism to specifically impose formation of hexameric lattices yet inhibit the formation of pentamer declinations that are needed to close off these lattices to form mature, conical capsids.

## LEN hyper-stabilizes WT and M66A CA assemblies during virion maturation

We previously demonstrated that LEN hyper-stabilized pre-assembled hexameric lattices of recombinant WT CA and pre-formed mature WT HIV-1 capsid in target cells and *in vitro* [14]. Here, we extended our studies by adding LEN to virus producer cells, and then examining the stability of the subsequently formed capsid *in vitro* using the CDR-labeling assay [30]. CDR, which stands for Cyclophilin A–DsRed, tightly engages capsid. Following virus membrane permeabilization with saponin, capsid disassembly or uncoating is tracked via CDR signal loss. The results in Fig 9 show that the inhibitor strongly stabilized WT HIV-1 capsid (Fig 9A and 9B). In parallel experiments, LEN also stabilized the HIV-1$_{(M66A\ CA)}$ capsid, albeit substantially higher concentrations of the inhibitor were needed to elicit these effects on the mutant vs. WT virus (Fig 9C and 9D). These results suggest that the M66A substitution confers substantial resistance to LEN, due presumably to the propensity of the mutant protein to preferentially form pentameric assemblies, which are not targeted by LEN [5,6]. Yet, higher LEN concentrations promoted formation of and hyper-stabilized the hexameric lattices of CA(M66A) *in vitro* and in virus producer cells (Figs 8 and 9).

## Discussion

We demonstrate that the antiviral potency of LEN during the late steps of HIV-1 replication in cell culture is strongly affected by the CA levels. LEN displayed $EC_{50}$ of ~ 60 pM at the lowest p24 levels (~ 583 pM) assayed here. In HIV-1 infected individuals, the average p24 levels can vary depending on the stage of infection, individual immune responses and other concurrent conditions [31–35]. The ultra-sensitive detection assay of p24 in 92 HIV-1 infected people identified median and highest levels of the viral antigen to be ~ 0.04 pM and ~ 8.3 pM, respectively [36], which are substantially lower than the p24 levels in our cell culture assays (Fig 1A). Since we observed an inverse correlation indicating that the LEN potency is substantially increased with decreasing levels of p24, it is logical to suggest that in the clinical setting LEN can potently (with pM $EC_{50}$s) inhibit HIV-1 maturation.

LEN is administered subcutaneously to HIV-1 infected people once every six months [13]. Plasma LEN concentrations peak at < 100 nM within a few days of the subcutaneous injection and then decrease substantially (< 10 nM) over several months [13]. Thus, the long-acting property of LEN relies on its pM antiviral activity [13]. Therefore, it is important for mechanistic and structural studies to focus on delineating the primary, highly potent antiviral mode of action of LEN from secondary effects that can be observed with artificially higher inhibitor concentrations.

A recent preprint reported cryo-EM structures of LEN bound to native immature HIV-1 particles, which revealed that the inhibitor altered the architecture of the Gag lattice [37]. However, this structural study used a concentration of inhibitor (300 μM) that greatly

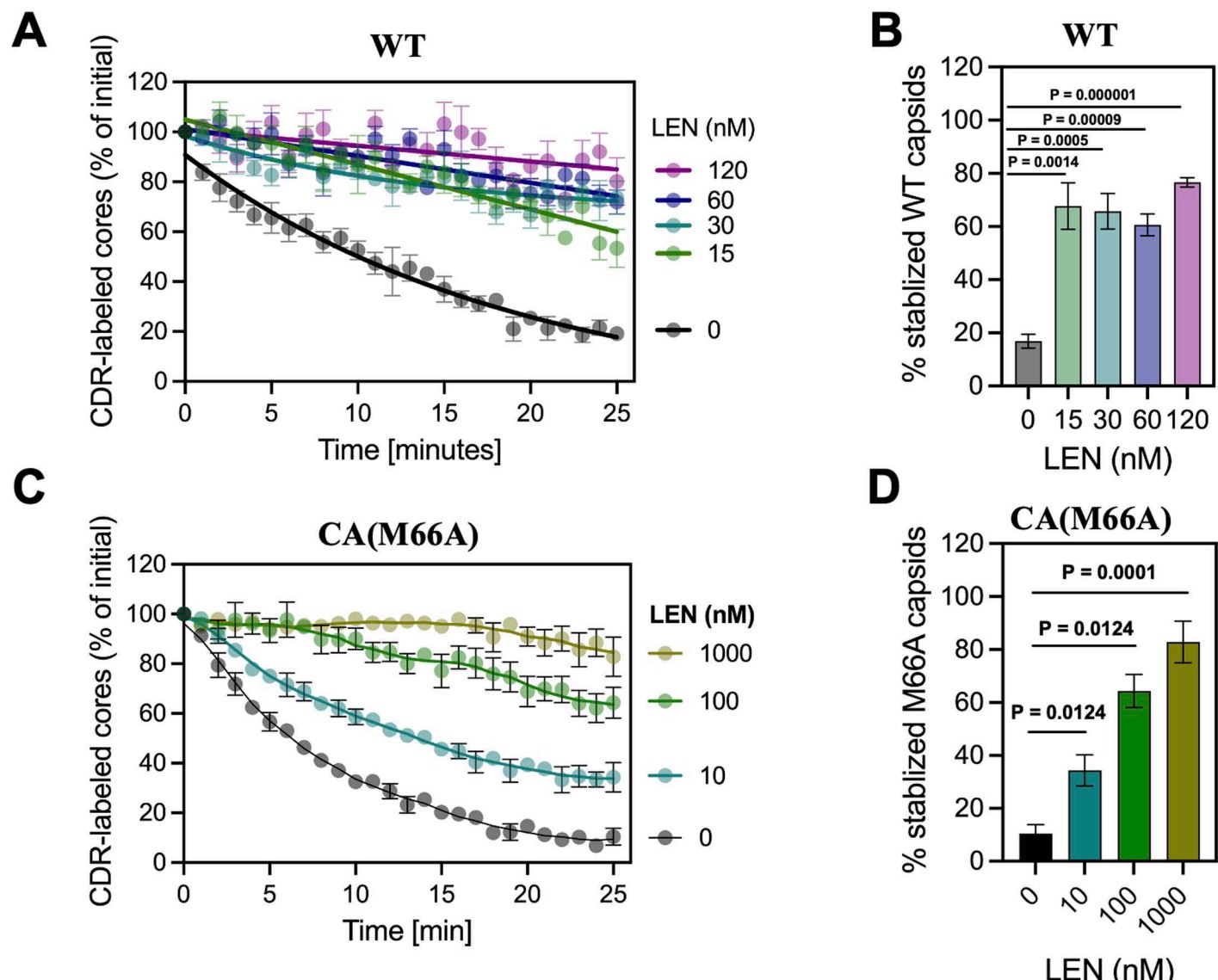

**Fig 9. HIV-1 capsids produced in the presence of LEN show increased stability *in vitro*.** (A and B) Quantification of the CDR/capsid marker loss from INmNG labeled WT HIV-1 capsids produced in the presence of DMSO (vehicle control) or indicated concentrations of LEN. Kinetics of CDR loss (A), and endpoint (25 min) measurements of stable HIV-1 capsid (B) are shown. (C and D) Quantification of the CDR/capsid marker loss from INmNG labeled HIV-1(M66A CA) capsids produced in the presence of DMSO (vehicle control) or indicated concentrations of LEN. Kinetics of CDR loss (C), and endpoint (25 min) measurements of stable HIV-1 capsid (D) are shown. Student 't'-test was used to determine statistical significance in (B and D).

exceeded (> 100,000-fold) what is required to inhibit HIV-1 replication under physiological conditions [37]. Therefore, it is unclear if the detected conformational changes in Gag would also occur at lower inhibitor concentrations, or whether altered Gag conformations do not affect the process of virus maturation. Indeed, the present study demonstrates that pharmacologically relevant LEN concentrations do not detectably affect Gag proteolytic processing during virion maturation (S3 Fig).

Instead, we uncovered the primary mechanism of action of LEN during virus particle morphogenesis. Treatments of virus producer cells with sub-stoichiometric LEN:CA ratios resulted in aberrant CA assemblies. LEN blocked the formation of pentamers whereas it

promoted formation of hexameric lattices. LEN induced an opened conformation in CA conducive to hexamer but not pentamer formation. Furthermore, LEN strongly stabilized WT CA dimers. This novel finding is consistent with the structural data [13,14] indicating that LEN must engage with two adjoining CA subunits – the NTD of one subunit and the CTD of the neighboring subunit – for high affinity binding. In turn, LEN stabilized dimers impeded formation of pentamers, which require an odd number of CA subunits with a closed conformation. By contrast, LEN stabilized CA dimers subsequently formed hexamers and tubular assemblies.

While Fig 10 offers a simplified model for inhibition of the mature capsid assembly by LEN, our TEM experiments in the presence of both IP6 and the inhibitor indicate more complex interplay of these small molecules with CA. We suggest that sub-stoichiometric LEN does not fully block formation of IP6 induced pentamers. Instead, a disbalance between hexamers and pentamers introduced by LEN is sufficient to impair the assembly of CLPs *in vitro* and mature capsids in virions. Indeed, *in vitro* assemblies of CA in the presence of both IP6 and LEN revealed evidence for curved structures, indicative that some pentamers were formed (Fig 4C). Yet, unlike CLPs (Fig 4A), these aberrant CA assemblies were not fully closed and tended to clump together (Fig 4C).

Akin to the biochemical assays, atypical capsid assemblies were predominant in virions produced in the presence of LEN (Fig 2). While our TEM analysis suggested that a fraction of LEN treated samples contained seemingly mature capsids, future higher resolution electron tomography studies are warranted to delineate LEN-induced gross deformations from more subtle alterations of capsid assembly (Fig 2). We would draw comparison with previous studies of allosteric integrase inhibitors (ALLINIs). While drug treatment quantitatively shifted

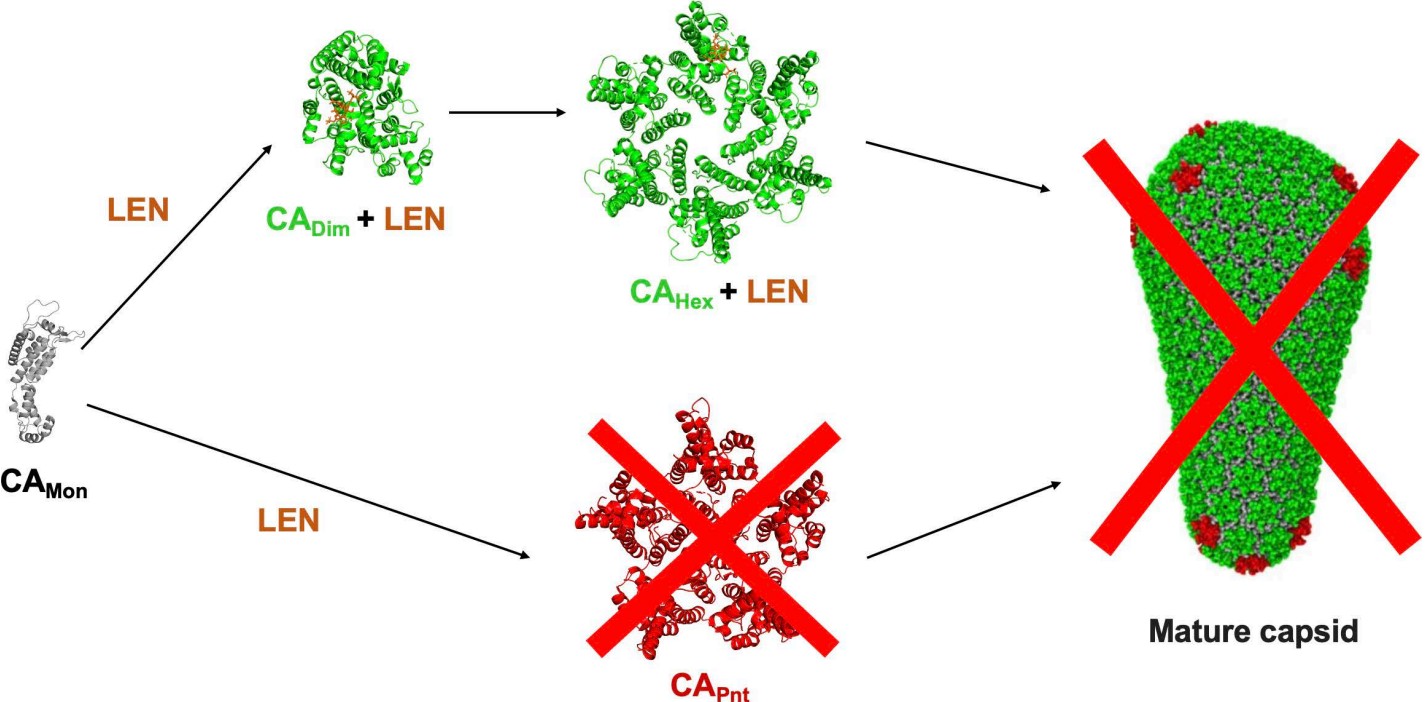

**Fig 10. A schematic for inhibition of mature capsid assembly by LEN.** Addition of LEN to CA stabilizes dimeric and hexameric intermediates, and specifically impairs formation of pentamers. More complex interplay in the presence of both IP6 and LEN with CA is expected as discussed in the text. $CA_{Mon}$, $CA_{Dim}$ + LEN, $CA_{Hex}$ + LEN (PDB id: 6VKV); $CA_{Pnt}$ (PDB id: 7URN); mature capsid (PDB id: 3J3Q).

mature particles to look eccentric with the viral ribonucleoprotein complex situated outside the capsid shell, minor viral fractions retained the WT phenotype by TEM [38]. Subsequent tomography studies revealed additional, more subtle capsid deformations induced by ALLINI treatments [39].

We demonstrate that sub-stoichiometric LEN:CA ratios suffice to form hyper-stable, malformed CA assemblies that are incapable of infecting target cells. This observation is consistent with our previous findings that the addition of pharmacologically relevant concentrations of LEN to target cells did not displace the cellular cofactor CPSF6 from HIV-1 capsids [40]. Given that tightly bound CPSF6 engages with only a subset of hydrophobic CA pockets targeted by LEN, sub-stoichiometric LEN can interact with unoccupied hydrophobic pockets and hyper-stabilize the capsid [40] (Figs 3 and 9). Future structural studies with sub-stoichiometric LEN bound to hexameric CA lattices are warranted to fully appreciate the atomic details of the highly potent antiviral mechanism of action of LEN and to leverage this information in future drug development efforts.

Taken together, the present study argues strongly for the clinical relevance for LEN to inhibit mature HIV-1 capsid assembly and extends our previous observations [14,40] by highlighting the following two distinct mechanisms for pM antiviral activities of the inhibitor: 1) during virus ingress LEN blocks the nuclear import and integration of HIV-1 by hyper-stabilizing the mature capsid, and 2) during virus egress LEN impairs the assembly of the mature capsid by exerting opposing effects on the formation of pentamers and hexamers, key assembly intermediates in mature capsid formation. These findings will inform clinical applications of LEN, emergence of drug-resistance phenotypes, and development of second-generation HIV-1 CA inhibitors.

## Materials and Methods

### Cells

HEK293T, HeLa, and TZM-bl cells were obtained from NIH AIDS Reference and Reagent Program and cultured in high-glucose Dulbecco's Modified Eagle Medium (DMEM, Mediatech, Manassas VA) supplemented with 10% fetal bovine serum (FBS, Sigma, St. Louis, MO) and 100 U/mL penicillin-streptomycin (PS, Gemini Bio-Products, Sacramento, CA). All the cell lines were cultured in the 37 °C incubator with 5% $CO_2$ and passaged regularly. Mycoplasma contamination was measured monthly using Mycoscope Mycoplasm PCR detection kit (Genlantis) and no contamination was detected.

### Virus production and antiviral assays

We performed late-stage antiviral activity assays for LEN or DRV based on our previously described protocols with some modifications [25,41]. LEN was prepared as described [14]. Briefly, HEK293T cells (producer cells) were seeded at concentrations 3–5 x $10^5$ cells/mL, 1 day prior to transfection with a range of DNA concentrations (0.03125 μg, 0.125 μg, 0.5 μg, and 2 μg) of HIV-1 NL4-3 full-length replication competent plasmid using X-tremeGENE HP (Roche Applied Science) transfection reagent as per manufacturer's instructions. The pUC18 plasmid (backbone empty plasmid) was used along with NL4.3 DNA to normalize total DNA levels across conditions. The medium was removed and replaced with fresh medium in the presence of the compounds at the indicated concentrations or DMSO (diluent control) at 4–6 h post-transfection followed by incubation at 37 °C. Virus-containing supernatant was collected at 48 h post-transfection, filtered through 0.45 μm filter, and stored at −80 °C until further processing. The cell pellet was collected to quantify the Gag protein by western blot.

For antiviral activity assays in Fig 1, 9 ml of virus containing supernatants were subjected to ultracentrifugation through 20% (w/v) sucrose cushions at 32,000 rpm, 4 °C for 2 h. The supernatants were removed without disturbing the pellet. Virus pellets were resuspended in 100 µl STE buffer (10 mM Tris-HCl [pH 7.4], 100 mM NaCl, 1 mM EDTA, 40 µM IP6) and subjected to centrifugation at 21,000 xg, 4 °C for 8 min. The supernatants were removed, and the virus pellets were dissolved in 200 µl PBS. P24 levels were measured by ELISA using XpressBio HIV-1 p24 ELISA Kit (Catalog# XB-1000). To quantify the LEN levels bound to the viruses, 100 µl virus containing solution was mixed with 300 µl Acetonitrile + Methanol (50:50 ACN:MeOH) and centrifuged at 14,000 xg for 5 min. The clear supernatants (10 µl) were analyzed by LC-MS/MS instrument as described below (see LC-MS/MS assays of LEN).

For infection, HeLa TZM-bl cells (target cells) were seeded at 50,000 cells/well (24 well plates) on day 0. On day 1, the viruses were added to the cells and incubated at 37 °C for 3–4 h before removing and replacing with fresh medium. At 48 h post-infection, cells were collected, and luciferase activity was measured. Half maximal effective concentration ($EC_{50}$) of LEN was determined by Origin 2019 (v9.6) software. Virus infections were done in the presence of 8 µg/mL polybrene, and values were expressed as mean ± standard deviation (SD).

For assays in Fig 3, 2 µg of HIV-1 NL4-3 was used to transfect HEK293T cells. The final virion preparations in 200 µl PBS obtained following pelleting through 20% (w/v) sucrose cushions (see above) were separated in three parts for 1) LC-MS/MS, 2) p24 ELISA, and 3) infectivity assays. 1) To quantify the LEN levels bound to virions, the virus containing solution (100 µl) was mixed with 300 µl acetonitrile + methanol (50:50 ACN:MeOH) and centrifuged at 14,000 xg for 5 min. The clear supernatants were analyzed by LC-MS/MS instrumentation as described below (see LC-MS/MS assays of LEN). 2) P24 levels were measured by sandwich ELISA using commercially available XpressBio HIV-1 p24 ELISA Kit (Catalog# XB-1000). The above assays allowed us to determine molar concentrations of LEN and CA (p24) present of the final virion preparations (in 200 µl PBS) and calculate the LEN:CA ratios. 3) The infectivity of these virions in TZM-bl cells was determined as described above.

## Fluorescent virus production

Fluorescently labeled viruses were produced in HEK293T cells by transient transfection of the following plasmids: (i) 2 µg pHIV backbone containing WT CA or CA(M66A); (ii) 0.8 µg pVpr-INmNeonGreen; and (iii) 0.5 µg plasmid encoding VSV-G envelope glycoproteins. Jet-Prime transfection reagent was used per manufacturers protocols. Where indicated, the plasmid encoding the capsid marker pCyclophilinA-DsRed (CDR, 0.8 µg) [30] was co-transfected to add a secondary capsid label to the virus for uncoating studies. After 6 h of transfection, the media was exchanged with a complete phenol-red (-) DMEM containing DMSO or indicated concentrations of LEN. After additional 36 h of incubation in a 37 °C $CO_2$ incubator, virus supernatants were collected by centrifugation and filtration through a 0.45 µm filter to remove cellular debris. All virus preparations were quantified by the product-enhanced reverse transcription (PERT) assay, aliquoted, and stored at −80 °C until further use.

## Capsid stability measurements *in vitro*

For *in vitro* capsid stability measurements, WT HIV-1 and HIV-1$_{(M66A\ CA)}$ pseudovirus particles, fluorescently labeled with the INmNG (viral ribonucleoprotein complex-marker) and CDR (capsid marker) were immobilized on poly-l-lysine treated 8-well chambered slide (#C8-1.5H-N, CellVis) by incubating at 4 °C for 30 min. Unbound virus particles and LEN in the supernatant were removed by 5x washes in PBS. The bound virus particles were imaged on a Leica SP8 LSCM, using a C-Apo 63x/1.4NA oil-immersion objective. Tile-scanning

was employed to image 2 x 2 neighboring fields of view. Saponin (100 μg/ml) was added to permeabilize the virus membrane and initiate capsid uncoating *in vitro*. Time-lapse images were collected at 1-min intervals for 30 min to visualize the loss of CDR from INmNG-labeled capsid. Cyclosporin A (CsA, 5 μM) was added at 25 min after permeabilization and onset of imaging to displace the CDR from the capsid that had not disassembled during the imaging time window.

## Image analyses

For *in vitro* capsid-stability measurements, INmNG and CDR puncta were detected using the wavelet spot detector in ICY. The kinetics of capsid disassembly *in vitro* were determined by normalizing the number of CDR puncta to that at t = 0 min and subtracting the background CsA-resistant CDR puncta corresponding to immature viruses at t = 25 min [30]. The total number of INmNG puncta in each field of view remained constant over the 25-min imaging period and served as a reference marker for particle displacement. Capsid stability was determined by plotting the loss of CDR spots over time, normalized to the initial number of spots. Immature particles that retained CDR and failed to respond to CsA treatment were excluded from the analysis.

## Transmission electron microscopy of virions

Viruses were generated by transfecting HEK293T cells ($10^7$ plated in 15 cm plates the previous day) with 30 μg pNL4-3 plasmid DNA using PolyJet DNA transfection reagent (SignaGen Laboratories) according to the manufacturer's protocol. LEN, where indicated, was added 5 h post-transfection by replacing the media with fresh DMEM containing the indicated drug concentration. LEN-containing media was prepared by making serial 1000 x stocks in DMSO, which was then added to the media. Drug-free media was accordingly adjusted to 0.1% final DMSO concentration. Two days post-transfection, virus-containing cell supernatants were clarified by centrifugation at 800 xg for 15 min and subsequently filtered by gravity flow through 0.45 μm filters. CA levels were assessed by p24 ELISA using a commercial kit (Advanced Bioscience Laboratories, Cat. #5447). Viruses were pelleted by ultracentrifugation using a Beckman SW32-Ti rotor at 26,000 rpm for 2.5 h at 4 °C. Virus pellets resuspended in 1 mL fixative solution (2.5% glutaraldehyde, 1.25% paraformaldehyde, 0.03% picric acid, 0.1 M sodium cacodylate, pH 7.4) were incubated at 4 °C overnight. The preparation and sectioning of fixed virus pellets were performed at the Harvard Medical School Electron Microscopy Core Facility as previously described [42]. Sections (50 nm) were imaged using a JEOL 1200EX transmission electron microscope operated at 80 kV. All images used for quantitative analysis were taken at 30,000X magnification. Images were uploaded to ImageJ and analyzed using the Cell Counter plugin, which allows for numbered markers corresponding to categories to be assigned to individual virions. Only virions that were within the plane of focus with an electron-dense outer membrane were considered for quantification. All virions were assigned one of three morphological categories: (1) mature,(2) immature, and (3) atypical. The characterization of mature and immature virions was based on previously published descriptions of these phenotypes [42]. All other categories were established based on qualitative observations made prior to image quantification. Examples and descriptions of each morphological category are provided in Fig 2. Within each experiment, we counted at least 200 virions per condition. The number of virions per category was summed and normalized as a percentage of total virions within that set. Unpaired Student's t-tests were performed in Prism Version 9.0.1 to assess differences in the mean frequencies for each morphological category as compared to the WT/no-drug control.

## LC-MS/MS assays of LEN

An Agilent 6495 (Agilent Technologies; Santa Clara, CA) equipped with an Agilent 1290 Infinity II HPLC and Autosampler were used for LC-MS/MS method development. Liquid chromatography employed a Waters Acquity BEH C8 2.1 x 50 mm 1.7 µM, 40 °C with a flow-rate of 0.4 mL/min. The mobile phase consisted of solvent A: 10 mM ($NH_4OAc$), 0.1% formic acid in $H_2O$; and solvent B: 45:45:10 ACN:IPA:$H_2$0 with 10 mM (NH4OAc) and 0.1% formic acid. For chromatography 20% B was increased to 100% B from 0 to 3.0 min and held for 1.0 min, then changed back to 20% B at 4.0 min and held for 1.0 min with the total run time of 5 min. The compound was monitored via electro-spray ionization positive ion mode (ESI+) using the following conditions: i) a capillary voltage of 4000 V; ii) Gas flow and temperature at 18 L/min and 250 °C; iii) Sheath gas set at 12 L/min and 350 °C; iv) Nebulizer pressure at 35psi; v) Nozzle charging voltage at 300 V; vi) quadruple one (Q1) and (Q3) were set on Unit resolution; vii) dwell time was set at 50 msec; and viii) Optimal collision energy (CE) for each compound was determined from flow injection analysis of authentic reference materials prepared at 1 µg/ml. From 10 mM LEN stock solution, working solutions were prepared at 100 µg/ml, then serially diluted to prepare standard curves. Method for LEN compound operating in positive ion mode was developed. Settings: (tR = 2.3 min) displayed 3 predominate MRM's transitions of the M+H adduct, 967.9 −> 868.6 m/z (Quantitative MRM), CE = 32; 967.9 −> 947.7 m/z (Confirmatory MRM #1), CE = 24, and 967.9 −> 888.5 m/z (Confirmatory MRM #2), CE = 36. The limit of detection was found to be 0.44 ng/mL.

## Preparation of recombinant proteins

WT and M66A CA recombinant proteins were expressed from pET3a-CA plasmids. CA(N21C/A22C/W184A/M185A), CA(A14C/E45C/W184A/M185A), and the NTD of CA were expressed from pET11a plasmids. Recombinant proteins were expressed from *E. coli* (BL21-DE3) cells and purified as previously described [22,43–45]. Full-length WT Gag and Gag(Δp6) were expressed in *E. coli* BL21(DE3) at 37 °C and lysed by sonication in 750 mM NaCl, 20 mM Tris (pH 7.5), 10 mM 2-mercaptoethanol (βME), 10% glycerol, 0.05% Triton X-100, 1 µM $ZnCl_2$. Soluble lysate fractions were treated with 0.15% polyethylenimine to precipitate endogenous nucleic acids prior to the ammonium sulfate precipitation step. The pellets were resuspended in 100 mM NaCl, 20 mM Tris-HCl (pH 7.5), 10 mM βME and 1 µM $ZnCl_2$, loaded onto a HiTrap Heparin HP affinity column (Cytiva) equilibrated in the same buffer, and eluted with the 0.1 − 1 M NaCl gradient. The peak fractions containing the protein of interest were concentrated and subjected to size exclusion chromatography (SEC) using HiLoad 16/600 Superdex 200 pg column (Cytiva) with a running buffer of 20 mM Tris-HCl (pH 7.5), 500 mM NaCl, 10 mM βME and 1 µM $ZnCl_2$. The peak fractions of full-length Gag and Gag(Δp6) were concentrated and flash frozen in liquid nitrogen.

## Formation of cross-linked CA pentamers and hexamers

Cross-linkable pentamer (CAN21C/A22C/W184A/M185A) and hexamer (CAA14C/E45C/W184A/M185A) proteins were purified as described [23]. The proteins were then diluted to 1 µM in buffer (50 mM Tris-HCl pH 8.0, 50 mM NaCl) and then separated into two groups: with LEN (2 µM final concentration) and control (with DMSO), and the proteins were then concentrated to 125 µM using Amicon Ultra 3K (Merck Millipore). The oligomers were obtained by sequential dialysis adapting the protocol [22], using 125 µM protein into first assembly buffer (50 mM Tris-HCl pH 8.0, 50 mM NaCl, 40 mM β-mercaptoethanol (βME) overnight, then aliquoted (100 µL) to mini dialysis units (Thermo Scientific) and put into second assembly buffer (50 mM Tris-HCl pH 8, 1 M NaCl) containing 100 mM βME for 2 h.

After that, the protein was moved to third assembly buffer with 0.2 mM βME for 2 h and, in the end, moved to a final 20 mM Tris-HCl pH 8, 40 mM NaCl buffer without βME. All dialysis steps were performed at 4 °C, with mild agitation. The fractions were then collected and analyzed through SDS-PAGE.

## Assembly of CLPs

Conical HIV-1 CLPs were assembled by incubating the purified, recombinant CA protein (3 mg/mL) with 500 µM or 10 mM IP6 (Sigma), 50 mM MES, pH 6, 40 mM NaCl, and DMSO at room temperature for 16 h. For LEN (MedChemExpress) treatment, CA protein (3 mg/mL) was incubated with 117 µM LEN, 50 mM MES, pH 6, and 40 mM NaCl at room temperature for 16 h. CA protein incubated with DMSO, 50 mM MES, pH 6, and 40 mM NaCl was used as a control assembly reaction. For LEN and IP6 co-treatment, the HIV-1 CLPs were assembled at 37 °C for 1 h by incubating with the indicated LEN concentrations (based on the molar ratio versus CA protein), 500 µM IP6, 50 mM MES, pH 6, and 40 mM NaCl buffer. CA protein assembly reactions were applied to glow-discharged formvar/carbon 300-mesh Cu grids for negative staining and TEM imaging.

## Negative staining transmission electron microscopy of CLPs

Assembly samples (10 µL) were applied to glow-discharged formvar/carbon 300-mesh Cu grids (Electron Microscopy Sciences, Hatfield, PA) and incubated for 2 min. After blotting, grids were washed for 2 min with 0.1 M KCl (for CA assembly products formed in the presence of IP6 and/or LEN) or 1 M NaCl (for the assembly products obtained in the high ionic strength buffer). Subsequent blotted grids were stained with 2% uranyl acetate for 2 min and blotted dry. Micrographs were collected by FEI Tecnai T12 Spirit (ThermoFisher, Waltham, MA) at 120 kV.

## Mass photometry

The reaction conditions described above (see Assembly of CLPs) were also used to monitor CA assembly intermediates by mass photometry. The measurements were carried out in silicone gaskets (3 mm × 1 mm, Grace Bio-Labs) on microscopy slides. The reaction solutions (2 µL) were added to the gaskets containing 18 µL buffer and images were acquired for 60 s at 331 Hz. The resulting data were analyzed using DiscoverMP (Refeyn Ltd) to extract particle contrasts. The calibration was made with Beta-amylase (BAM) and the analysis to obtain the gaussians and corresponding MWs were performed with the built-in tools in the software.

## Surface plasmon resonance

Surface plasmon resonance (SPR) experiments were performed using the Reichert 4-SPR. A nitrilotriacetic acid (NTA) sensor chip (HC200M, XanTec) was conditioned with 40 mM $NiSO_4$ at a flow rate of 50 µL/min for 3 min. Full-length Gag was immobilized on the NTA sensor chip via the C-terminal His-tag. The running buffer was HBS-P + (Cytiva) diluted 10 times to a final concentration of 0.01 M HEPES pH 7.4, 0.15 M NaCl, 0.05% v/v Surfactant P20 plus 5% DMSO. LEN was prepared by serial dilution in DMSO and then further diluted in the buffer to final 5% DMSO. The LEN concentrations varied between 4 µM and 31 nM. A recovery step was included into the program with the sensor chip being regenerated with 350 mM EDTA and 50 mM NaOH. For each interaction, background binding and drift were subtracted via a reference surface. The experiments were performed in triplicate and data was analyzed using TraceDrawer 2.0 and fit with a simple kinetic model with a term for mass transport when necessary.

## Proteolytic processing of recombinant HIV-1 Gag proteins

2 μM LEN or DMSO were added to 1 μM purified full-length Gag or Gag(Δp6), and then proteolytic reactions were initiated by addition of 25 ng recombinant HIV-1 protease (Abcam ab215523). The reaction buffer contained 20 mM Tris-HCl (pH 6.8), 1 mM EDTA, 1 mM DTT, 0.1% Triton X-100, 10% glycerol. The reactions were stopped by adding the SDS-PAGE sample buffer and boiling immediately. The cleavage products were visualized by SDS-PAGE and staining with Acquastain (Bulldog Bio).

## X-ray crystallography

The NTD of CA was prepared and purified as described [27,45]. Protein was concentrated to 50 mg/mL and LEN was added to 1 mM from a 20 mM DMSO stock solution. After standing for 1 hour, sitting drop crystallizations were set up consisting of 2 μl of protein and 1 μl of well solution using Crystal Screen Cryo I kit (Hampton Research). Crystals grew after several weeks at 17 °C in a specific condition: 0.17 M Sodium acetate trihydrate, 0.085 M Sodium cacodylate trihydrate, 25.5% w/v Polyethylene glycol 8,000, 15% v/v Glycerol (pH 6.5). Crystals were briefly transferred to a cryo-protectant consisting of well solution supplemented with 25% glycerol, then flash-cooled in liquid nitrogen. The softwares XDS [46], Aimless [47], BALBES [48] and phenix.phaser [49] were used for data processing. The structure was determined via molecular replacement with using a HIV-1 CA NTD in complex with PF-3450074 (PDB ID: 2XDE) [28] as the search model. Refinement used REFMAC [50] and phenix.refine [51]. Model building was done with Coot [52] and the table was generated with Phenix (Macromolecular structure determination using X-rays, neutrons and electrons: recent developments in Phenix [53]. Data collection and refinement statistics are shown in S1 Table, and coordinates are deposited in the Protein Data Bank under accession code 8V23.

## Modelling of capsid pentamer and comparison with cross-linked crystallographic structures

The amino acid sequence of WT HIV-1 CA in FASTA format was used as input into Colab-Fold2 [54] to build a pentamer structure. The modelling was done with and without Amber relaxation step, with no significant differences observed. The cif file was then opened in PyMol (Schrodinger) along with cross-linked pentamer structure (PDB id: 3PO5). The command cealign was used to superimpose the two structures and obtain the corresponding RMSD.

## Supporting information

**S1 Table. Production of HIV-1 particles.**
(DOCX)

**S2 Table. Data collection and refinement statistics.**
(DOCX)

**S1 Fig. Antiviral activity of DRV during late steps of HIV-1 replication.** HIV-1 virions were produced by transfecting indicated amounts of the full-length, WT HIV-1NL4.3 plasmid in HEK293T cells (producer cells). Indicated concentrations of DRV or DMSO control were added to HEK293T cells, the excess DRV was removed by the Lenti-X concentrator, and the virions were used to infect HeLa TZM-bl cells (Target cells). After 48 h of infection, luciferase activity was measured to determine the EC50 values for DRV. The averaged data (+/−SD) from three independent experiments are shown.
(TIFF)

**S2 Fig. Effects of LEN on intracellular Gag levels.** HEK293T cells were transfected with 2 µg full-length, WT HIV-1$_{NL4.3}$ plasmid. After 4 h, increasing concentrations of LEN were added to the virus producer cells. After 48 h, cells were collected, and immunoblotting analysis was performed using anti-HIV1 p55 + p24 + p17 antibody (ab63917). GAPDH was used for internal control. Lane 1: molecular weight markers; lane 2: 2025 nM LEN; lane 3: 675 nM LEN; lane 4: 225 nM LEN; lane 5: 75 nM LEN; lane 6: 25 nM LEN; lane 7: 8.3 nM LEN; lane 8: 2.8 nM LEN; lane 9: DMSO control. The immunoblot is representative of results observed in two independent experiments.
(TIFF)

**S3 Fig. LEN does not affect Gag proteolytic processing during HIV-1 particle maturation.** (A) HEK293T cells were transfected with 0.125 µg full-length WT HIV-1 NL4.3 plasmid. After 6 h post transfection, the medium was removed and 1 nM LEN or DMSO control containing media were added to the virus producer cells. The supernatants containing viruses were collected at 0 h, 16 h, 24 h, 40 h, 48 h, ultracentrifuged, and analyzed by immunoblotting. (B) HEK293T cells were transfected with 2 µg full-length WT HIV-1 NL4.3 plasmid. After 30 h post transfection, the medium was removed and 50 nM LEN or DMSO containing media were added to the virus producer cells. The supernatants containing virions were collected at 0 h, 2 h, 4 h, 8 h, ultracentrifuged, and analyzed by immunoblotting. Representative images of at least three independent experiments are shown in A and B. The observed Gag proteolytic processing products including CA, matrix (MA) and nucleocapsid (NC) are indicated.
(TIFF)

**S4 Fig. Binding of LEN to recombinant full-length Gag.** (A) Representative SPR sensorgrams of LEN binding to full-length Gag showing association (crescent curve) and dissociation (decrescent curve); $K_D$, $k_{on}$ and $k_{off}$ values are indicated. (B) The $K_D$ value is determined using the Hill fit for (A).
(TIFF)

**S5 Fig. Proteolytic processing of recombinant Gag by HIV-1 protease.** (A) HIV-1 protease mediated cleavage of full-length, recombinant Gag (1 µM) was performed in the presence of 2 µM LEN (+) or DMSO control (−). Lane 1: input of full-length Gag. Lanes 2–11: reaction products. Lane 12: MW markers. (B) HIV-1 protease mediated cleavage of 1 µM Gag(Δp6) was performed in the presence of 2 µM LEN (+) or DMSO control (−). Lane 1: input of Gag(Δp6). Lanes 2–11: reaction products. Lane 12: MW markers. Representative SDS-PAGE images of at least three independent reactions are shown. The proteolytic products including the bands corresponding to p41, CA, MA and NC are indicated.
(TIFF)

**S6 Fig. Comparison of modelled WT CA pentamer with the cross-linked CA pentamer.** Superimposition of the AlphaFold2 generated CA pentamer structure (orange) and X-ray crystallographic structure of the cross-linked pentamer (PDB id: 3PO5, in purple); (A) top and (B) side views.
(TIFF)

**S7 Fig. Comparative analysis of the opened and closed conformations of CA NTD.** The distances between Cα of G60 and Cε of M66 are shown to delineate the opened and closed conformations. (A) NTD$_{LEN}$ (PDB: 8V23, present work); (B) CA$_{Hex}$ (PDB: 7URN); (C) CA$_{Hex}$ + LEN (PDB 6VKV); (D) NTD + PF74 (PDB: 2XDE); (E) NTD$_{Apo}$ (PDB: 5HGK); (F) CA$_{Pnt}$ (PDB: 7URN); (G) NTD + CAP-1 (PDB: 2JPR); (H) NTD + 1F6 (PDB: 4INB). Colored

protein segments indicate connecting helices H3 and H4. Black dash lines indicate distances between Cα of Gly60 and Cε of Met66.
(TIFF)

**S8 Fig. LEN binding is compatible with the opened but not the closed conformation of CA.** Structure of $CA_{Hex}$ (Green) + LEN (dark blue) (PDB: 6VKV) superimposed onto (A) NTD + LEN ($NTD_{LEN}$, steel blue, PDB: 8V23, present work); (B) $CA_{Pnt}$ (scarlet, PDB: 7URN); and (C) $NTD_{Apo}$ (magenta, PDB: 5HGK). Different conformations of Met66 side chains are shown. In panel A, the Met66 side chain is compatible with LEN binding to the opened conformation seen in $NTD_{LEN}$. In contrast, the Met66 side chains in native $CA_{Pnt}$ (B) and $NTD_{Apo}$ (C) encounters steric hindrance with respect to LEN.
(TIFF)

## Acknowledgments

We are grateful to Garry Morgan and Anza Darehshouri for the technical suggestions for the electron micrographs. Electron micrographs were collected at Boulder Electron Microscopy Services at the University of Colorado, Electron Microscopy Core Facility at the University of Colorado Anschutz Medical Campus, and Harvard Medical School Electron Microscopy Core Facility. We thank Jay Nix at ALS Beamline 4.2.2 for acquiring data for the crystal structure.

## Author contributions

**Conceptualization:** Szu-Wei Huang, Ashwanth C. Francis, Alan N. Engelman, Mamuka Kvaratskhelia.

**Funding acquisition:** Szu-Wei Huang, Ashwanth C. Francis, Alan N. Engelman, Mamuka Kvaratskhelia.

**Investigation:** Szu-Wei Huang, Lorenzo Briganti, Arun S. Annamalai, Juliet Greenwood, Nikoloz Shkriabai, Reed Haney, Michael L. Armstrong, Michael F. Wempe, Satya Prakash Singh, Ashwanth C. Francis.

**Methodology:** Michael L. Armstrong.

**Supervision:** Ashwanth C. Francis, Alan N. Engelman, Mamuka Kvaratskhelia.

**Writing – original draft:** Szu-Wei Huang, Alan N. Engelman, Mamuka Kvaratskhelia.

**Writing – review & editing:** Szu-Wei Huang, Lorenzo Briganti, Arun S. Annamalai, Juliet Greenwood, Nikoloz Shkriabai, Michael L. Armstrong, Michael F. Wempe, Ashwanth C. Francis, Mamuka Kvaratskhelia.

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
