## [Decision Letter · Decision Letter 0]

9 Jun 2024

Dear Professor Kvaratskhelia,

Thank you very much for submitting your manuscript "The mechanism for highly potent inhibition of mature HIV-1 capsid assembly by lenacapavir" for consideration at PLOS Pathogens. As with all papers reviewed by the journal, your manuscript was reviewed by members of the editorial board and by several independent reviewers. In light of the reviews (below this email), we would like to invite the resubmission of a significantly-revised version that takes into account the reviewers' comments.

Dear Dr. Kvaratskhelia,

I think this is a strong study. In the reviews attached, the reviewers make a number of comments that I think are reasonable to address. My thoughts on their comments having reviewed your paper again after receiving the reviews:

It seems like 2 reviewers are concerned with issues of gag maturation/binding to immature Gag, and how that might be impacting infectivity and other downstream results in figures not using recombinant CA. I think your data strongly indicate a defect in pentamer formation, and you seem to have chosen a drug concentration where Gag expression is not significantly affected in cell experiments, but a more robust characterization of gag processing and LEN binding to immature Gag seems warranted to appease many of the reviewer concerns in a resubmission. It seems like your approaches afford you a number of opportunities to address these points more fully. On the other hand, I cant myself conceive of how you might address the issue of assembly kinetics, but believe if you could, that would be an exceptional addition to the manuscript.

Addressing the point of M66A infectivity/LEN resistance seems feasible if no other study has examined this, but more importantly I think it is important to clarify and contextualize your results and interpretations with this and the Cyteine crosslinking CA mutants. Both are tools used by the field that facilitated you asking questions about pentamers that could not have otherwise been approached, but both have caveats related to their relevance. The fact that they both supported a similar conclusion, which is also supported by the studies of wt CA in figure S4. However, since the caveats noted by each reviewer are, to the best of my knowledge, fair and accurate, acknowledging these caveats in the discussion seems warranted, as these details may be lost on non-CA experts. Personally, I also am not sure why S4 is not a primary figure as currently designed, and this also seems like a great opportunity to address the stoichometry issue, but I am not familiar enough with this system to know if this is feasible.

We cannot make any decision about publication until we have seen the revised manuscript and your response to the reviewers' comments. Your revised manuscript is also likely to be sent to reviewers for further evaluation.

Sincerely,

Edward M Campbell, PhD

Academic Editor

PLOS Pathogens

Susan Ross

Section Editor

PLOS Pathogens

Michael Malim

Editor-in-Chief

PLOS Pathogens

orcid.org/0000-0002-7699-2064

Dear Dr. Kvaratskhelia,

I think this is a strong study. In the reviews attached, the reviewers make a number of comments that I think are reasonable to address. My thoughts on their comments having reviewed your paper again after receiving the reviews:

It seems like 2 reviewers are concerned with issues of gag maturation/binding to immature Gag, and how that might be impacting infectivity and other downstream results in figures not using recombinant CA. I think your data strongly indicate a defect in pentamer formation, and you seem to have chosen a drug concentration where Gag expression is not significantly affected in cell experiments, but a more robust characterization of gag processing and LEN binding to immature Gag seems warranted to appease many of the reviewer concerns in a resubmission. It seems like your approaches afford you a number of opportunities to address these points more fully. On the other hand, I cant myself conceive of how you might address the issue of assembly kinetics, but believe if you could, that would be an exceptional addition to the manuscript.

Addressing the point of M66A infectivity/LEN resistance seems feasible if no other study has examined this, but more importantly I think it is important to clarify and contextualize your results and interpretations with this and the Cyteine crosslinking CA mutants. Both are tools used by the field that facilitated you asking questions about pentamers that could not have otherwise been approached, but both have caveats related to their relevance. The fact that they both supported a similar conclusion, which is also supported by the studies of wt CA in figure S4. However, since the caveats noted by each reviewer are, to the best of my knowledge, fair and accurate, acknowledging these caveats in the discussion seems warranted, as these details may be lost on non-CA experts. Personally, I also am not sure why S4 is not a primary figure as currently designed, and this also seems like a great opportunity to address the stoichometry issue, but I am not familiar enough with this system to know if this is feasible.

Reviewer's Responses to Questions

**Part I - Summary**

Reviewer #1: In this manuscript, Huang et al. investigate the mechanism of inhibition of HIV-1 capsid assembly by lenacapavir (LEN) as previous reports have focused on the activity of LEN during the early stages of infection. Their investigation reveals several key features of LEN activity relevant for both clinical usage and for understanding of HIV-1 biology. Their data shows a clear dependence of the antiviral potency of LEN on p24 as compared to the protease inhibitor Darunavir, and further demonstrate convincingly that sub-stoichiometric LEN:CA ratios are sufficient to result in malformed capsids. Finally, they determined that aberrant capsid morphology results from LEN preventing pentamer formation, while promoting hexamer formation by forcing the capsid into an “open” confirmation. The determination of the novel mechanism by which LEN inhibits mature capsid assembly is important, and the conclusions drawn are fully supported by the data presented. Overall, the manuscript is clear, however I would strongly suggest that the supplemental figures concerning LEN and capsid assemblies (S4 and S3 Fig), and the overall model (S7 Fig) should be moved into the main text (or combined with the main figures), as they are crucial to the conclusions of the paper.

Reviewer #2: Huang and coworkers examine the mechanism by which the capsid-targeting inhibitor Lenacapavir reduces infectivity of HIV-1 particles that are released from cells cultured in its presence. They show that: (1) the “producer-cell” antiviral potency depends on the quantity of proviral plasmid used for transfection; (2) Len results in altered CA assembly products formed in vitro; virions formed in the presence of Len contain substoichiometric ratios of Len to CA; (3) Len inhibits the formation of crosslinked pentamers but not hexamers during assembly of mutant CA proteins in vitro; (4) Len affects the structure of CA-NTD protein in crystals formed in its presence; (5) CA-M66I assembles into tube rather than spheres in the presence of Len. They conclude that Len perturbs maturation by interfering with proper pentamer formation.

While the approaches and results are interesting, the interpretation seems overdone. Maturation is a kinetic process that involves cleavage of sites in Gag and Gag-Pol at different rates, disassembly of the immature lattice and assembly of the mature lattice, and closure of the capsid via pentamer formation. How pentamers form during HIV-1 capsid assembly is unknown. Since the authors performed no kinetic analyses of the effects of Len on lattice formation, they have ignored a possible simple explanation for the apparent impairment in biochemical, structural and/or architectural maturation. More importantly, some of the results may have alternative interpretations.

Reviewer #3: This study by Huang et al. describes the effects of lenacapavir (LEN) effects into assembly of HIV-1 capsids. The study highlights the effects of LEN on CA hexamers and how they affect the assembly due to the lack of correct pentamer formation required for the curvature of the capsid lattice.

The study is of significance to the field as LEN is a new treatment for people living with HIV and the study of mechanisms might have an impact into future therapy adaptations and predict escape mutations in the viruses. The structural and functional data presented is also very important to determine mechanisms of action of LEN and HIV viral function.

Strengths:

- LEN efficacy inversely correlates with p24 amount in producing cells, the results point that a higher amount of LEN molecules bound to CA result to a higher drug efficiency.

- Showing that LEN allows for the formation of dissulfide hexamers but not pentamers is a strong dataset for the premise of the article.

Weaknesses

- The experiments measuring the structure and effect of LEN on capsid assembly are only performed in CA monomers and do not take into consideration the effects of LEN into the Gag precursor and maturation. The authors should discuss these limitations, the caveats, and what could differ in maturation from Gag to CA to capsid lattice formation.

- The lack of experiments in primary cells that are targets to HIV-1 reduces the significance of the study.

**Part II – Major Issues: Key Experiments Required for Acceptance**

Reviewer #1: None

Reviewer #2: 1. The authors employ Cys mutants that stabilize hexameric and pentameric capsomeres by disulfide bond formation. They observed no effect on formation of crosslinked hexamers but inhibition of crosslinked pentamers. Notwithstanding the risk associated with interpreting results of Len on assembly of mutant proteins in vitro with respect to maturation of wild type HIV-1, it has been demonstrated that the structure of the pentamers formed by A21C/A22C mutant CA protein is markedly different from that of the pentamer present in native HIV-1 cores (PMID: 27980210). Thus, the relevance of the observed effect of Len on pentamer formation in this experiment may not be relevant to Len's effects on HIV-1 maturation.

2. Fig. 2: Actual Len:CA ratios in virions may have been reduced by removal of Len during virus purification and/or by loss during extraction. Therefore, the conclusion that “sub-stoichiometric LEN:CA ratios suffice to inhibit late steps of HIV-1 replication” (line 141) may be erroneous. It remains possible that the antiviral mechanism involves stoichiometric association, since Len may preferentially bind to the assembled CA in the virion vs. the CA protein that is not part of the capsid lattice, which is thought to be approximately half of the total CA in mature virions.

3. Fig. 4: the authors used CA protein with the M66A substitution for these experiments. The Ile substitution at this position is known to confer Len resistance and a large HIV-1 fitness cost, so the relevance of the approach seems doubtful. Does M66A also confer Len resistance? If so, how do the authors explain the effect of Len?

4. The immunoblot shown in S2 Fig is highly suggestive of impaired cleavage in Gag. This should be fully characterized, ideally through pulse-chase assays of Gag processing in infected cells and virions. Impaired Gag cleavage could well be the basis for the observed maturation defect engendered by Len.

Reviewer #3: - The usage of Lenti-X and the LEN found in supernatants without viruses decrease the excitement of the manuscript. The 3~4x lower trace amount of LEN that is found in supernatants without viruses could still represent an action in target cells and not only in the production stage. This is important as LEN has such a potency at pM doses. A sucrose cushion would result in a cleaner approach to remove LEN from the SN (as the volume of the cushion would prevent contamination, unlike using the PEG approach were there's no solution in between the phases). As an alternative so that to not perform the whole study with that methodology, it is advisable to provide LC/MS-MS data with LEN with a retrovirus known not to bind to LEN (such as MLV or HTLV). Further, infections in TZMbl using the same concentrations of LEN that are measured by LC-MS/MS would attenuate this concern.

- The Xray crystalography data is performed in CA monomers and the experiments with mCLPs are also performed with "pre-matured" CA mixed with LEN/IP6/others. LEN and IP6 might bind to the forming Gag precursor and this experimental model does not take into consideration the presence of LEN and IP6 at the moment of maturation. LEN binding to the Gag precursor is likely to have different interaction profiles with the different a.a. in CA.

- From the data from Fig 1 and others, could the authors estimate how many LEN molecules per capsid would be sufficient to cause a maturation defect?

**Part III – Minor Issues: Editorial and Data Presentation Modifications**

Reviewer #1: 1) Lines 265-268: The authors note that hexamer field for CA(M66A) is lower than WT, but no data is shown in Fig.5 regarding yield. Is it possible to quantify the yields?

2) S1 Fig. – The figure legend states LEN was removed; this appears to be an error since this figure shows DRV treatment.

3) S3 Fig. - Panel and axis labels are missing for B and C.

4) I suggest moving S3 and (especially) S4 Fig to the main text, perhaps combining with Fig. 3.

5) The model (S7 Fig) should also be included in the main body of the report.

Reviewer #2: 5. Line 150: “Gag degradation…” This has not been demonstrated. The apparent decrease in Gag accumulation in transfected cells could have another explanation, such as decreased translation, accelerated budding, or reduced solubility.

6. Discussion lines 281-285: the level of p24 in patient plasma seems irrelevant to the matter, as it likely has little to do with the concentration of Gag in a productively infected cell. The important comparison would be the Gag concentrations in their transfected cells vs. those in infected T cells. On that note, adding a producer cell Len titration experiment using infected T cells (even a T cell line) could increase confidence in the relevance of the data.

Reviewer #3: Line 56-58 - "The integrity of the mature capsid architecture is important for protecting the viral genome from immune sensing during subsequent trafficking of HIV-1 cores throughout the cytoplasm to successfully reach the nucleus of infected target cells."

The opposite reasoning has also been shown: Only an intact capsid is targeted by TRIM5 and Mx2, which points to the opposite hypothesis.

As the matter is still under debate and it's a model among many for the early steps of infection, the authors could revise the sentence.

Line 70- The Study by the Diaz Griffero lab (PMID35005542) claims that LEN/GS-CA1 allows for nuclear import, the authors should discuss the differences and what could explain the discrepancy of the results.

Related to Line 87 and the discussion - The authors should further discuss the biological relevance of the findings as it is unknown how many capsids are produced per cell at a given moment, in different cell types.

Line 134/Fig2.B. The concentrations are shown in ng/mL while the rest of the manuscript shows molarity. Please convert or add the information in nM/pM for reading clarity purposes.

Line 149-150. The figure showing p24 vs p55(Gag) show less Gag in the higher >75nM LEN. It is also visible a stronger p24 band. The authors should discuss this finding. The authors claim Gag "degradation", perhaps another word would be more appropriate, as no degradation studies were conducted and could be simply a difference into the protease activity efficiency.

Fig S3. B and C labels are absent.

Fig S7 model should be improved in relation with the structural findings shown in the results.

Lenacapavir source should be stated in material and methods. (and please confirm that it's lenacapavir GS-6207 and not GS-CA1).

PLOS authors have the option to publish the peer review history of their article (what does this mean? ). If published, this will include your full peer review and any attached files.

**Do you want your identity to be public for this peer review?** For information about this choice, including consent withdrawal, please see our Privacy Policy .

Reviewer #1: No

Reviewer #2: No

Reviewer #3: No
---

## [Editor Report · Decision Letter 1]

19 Dec 2024

PPATHOGENS-D-24-00850R1

The primary mechanism for highly potent inhibition of HIV-1 maturation by lenacapavir

PLOS Pathogens

Dear Dr. Kvaratskhelia,

Thank you for submitting your manuscript to PLOS Pathogens. The editor handling the study has reviewed your revised manuscript and finds it to be much improved and responsive to reviewer comments. However, a few issues were noted, particularly with data new to this revised manuscript, that should be addressed. Therefore, we invite you to submit a revised version of the manuscript that addresses the points raised during the review process.

Please submit your revised manuscript within 30 days Feb 17 2025 11:59PM. If you will need more time than this to complete your revisions, please reply to this message or contact the journal office at plospathogens@plos.org. Please include the following items when submitting your revised manuscript:

We look forward to receiving your revised manuscript.

Kind regards,

Edward M Campbell, PhD

Academic Editor

PLOS Pathogens

Susan Ross

Section Editor

PLOS Pathogens

Michael Malim

Editor-in-Chief

PLOS Pathogens

orcid.org/0000-0002-7699-2064

**Additional Editor Comments:**

The mass photometry in Figure 6 is impressive. However, the text describes data in the presence of LEN after 1 hour, but 10 min and 16h are shown? Assuming all panels are accurately labelled, since the data in D/E show 10 min/1hr in presence of IP6, it seems useful to compare both conditions at 1 hr and 16hr. Based on the text, one might infer this has been done, so I would consider this an editorial revision.

Please describe in the text how you evaluated p24 levels, measured by ELISA, to infer the number of CA molecules and in turn, what this means about the molecules of LEN that can affect virion infectivity, given that a substantial amount of CA is not assembled into the core during viral maturation. This will both clarify the utility of this analysis to readers outside of our field but also provide interesting context for readers more familiar with the topic.

Please reconsider the use of “excessively high concentrations” in the discussion describing reference 35. Acknowledging that it is much higher than physiological concentrations seems quite appropriate, but “excessively high” could be viewed as unnecessary editorializing.

Despite it being a supplementary figure, S2 would really be much nicer and accessible if the LEN concentrations were included in the actual figure rather than the legend.

**Journal Requirements:**

Please ensure that the funders and grant numbers match between the Financial Disclosure field and the Funding Information tab in your submission form. Note that the funders must be provided in the same order in both places as well.

**Reviewers' Comments:**

**Figure resubmission:**
---

## [Editor Report · Decision Letter 2]

27 Dec 2024

Dear Professor Kvaratskhelia,

We are pleased to inform you that your manuscript 'The primary mechanism for highly potent inhibition of HIV-1 maturation by lenacapavir' has been provisionally accepted for publication in PLOS Pathogens.

Best regards,

Edward M Campbell, PhD

Academic Editor

PLOS Pathogens

Susan Ross

Section Editor

PLOS Pathogens

Sumita Bhaduri-McIntosh

Editor-in-Chief

PLOS Pathogens

orcid.org/0000-0003-2946-9497

Michael Malim

Editor-in-Chief

PLOS Pathogens

orcid.org/0000-0002-7699-2064
---

## [Editor Report · Acceptance letter]

Dear Professor Kvaratskhelia,

We are delighted to inform you that your manuscript, "The primary mechanism for highly potent inhibition of HIV-1 maturation by lenacapavir," has been formally accepted for publication in PLOS Pathogens.

Best regards,

Sumita Bhaduri-McIntosh

Editor-in-Chief

PLOS Pathogens

orcid.org/0000-0003-2946-9497

Michael Malim

Editor-in-Chief

PLOS Pathogens

orcid.org/0000-0002-7699-2064